# D-serine released by astrocytes in brainstem regulates breathing response to CO$_2$ levels

S. Beltrán-Castillo[1], M.J. Olivares [1], R.A. Contreras[1], G. Zúñiga[1], I. Llona[1], R. von Bernhardi [2] & J.L. Eugenín [1]

Central chemoreception is essential for adjusting breathing to physiological demands, and for maintaining CO$_2$ and pH homeostasis in the brain. CO$_2$-induced ATP release from brainstem astrocytes stimulates breathing. NMDA receptor (NMDAR) antagonism reduces the CO$_2$-induced hyperventilation by unknown mechanisms. Here we show that astrocytes in the mouse caudal medullary brainstem can synthesize, store, and release D-serine, an agonist for the glycine-binding site of the NMDAR, in response to elevated CO$_2$ levels. We show that systemic and raphe nucleus D-serine administration to awake, unrestrained mice increases the respiratory frequency. Application of D-serine to brainstem slices also increases respiratory frequency, which was prevented by NMDAR blockade. Inhibition of D-serine synthesis, enzymatic degradation of D-serine, or the sodium fluoroacetate-induced impairment of astrocyte functions decrease the basal respiratory frequency and the CO$_2$-induced respiratory response in vivo and in vitro. Our findings suggest that astrocytic release of D-serine may account for the glutamatergic contribution to central chemoreception.

[1] Departamento de Biología, Facultad de Química y Biología, Universidad de Santiago de Chile, USACH, Santiago 9170022, Chile. [2] Departamento de Neurología, Facultad de Medicina, Pontificia Universidad Católica de Chile, Santiago 8330024, Chile. Correspondence and requests for materials should be addressed to R.v.B. (email: rvonb@med.puc.cl) or to J.L.E. (email: jaime.eugenin@usach.cl)

Adjustment of breathing to physiological demands depends on central chemoreception, a sensory modality distributed through several brainstem nuclei that is also important for brain $CO_2$ and $H^+$ homeostasis[1, 2]. Central chemoreception fails in such life-threatening human conditions as central congenital hypoventilation syndrome, Rett syndrome, and sudden infant death syndrome (SIDS)[1]. ATP is released from the ventral medullary surface in response to $CO_2$ and $[H^+]$[3] and contributes to the activation of the respiratory network[3, 4]. Astrocytes can sense $CO_2$ and $H^+$ in neocortex[5, 6] and act as chemosensory interoceptors in the rat rostral medullary brainstem[4, 7] where they release ATP[4, 7] to activate $H^+$-sensitive retrotrapezoid neurons[4, 8–10] in response to acidosis or hypercapnia (increased levels of $CO_2$)[4]. However, experimental results supporting the theory that hypercapnia in the caudal medullary brainstem is also mediated by ATP have been elusive[11]. Differences either in astrocyte sensitivities to $H^+$ and $CO_2$, like those between rostral brainstem and cortical astrocytes[12], or in the release of gliotransmitters may account for regional discrepancies.

Neurons and astrocytes can synthesize and degrade D-serine[13–15], hence the presence of D-serine in various brain regions[13, 16, 17], including the brainstem. D-serine is an endogenous agonist with high affinity for the glycine-binding site of the N-methyl-D-aspartate (NMDA) glutamate receptor (NMDAR)[18–20]. D-serine distributes in the brain with the NMDAR[21]. At the cellular level, the endogenous co-agonist of synaptic NMDARs is D-serine, whereas the endogenous co-agonist of extrasynaptic NMDARs is glycine[22]. Although activation of the NMDAR is not essential for the generation of the respiratory rhythm[23–25], it does increase the respiratory frequency (fR)[23, 25]. Glutamate receptor activation contributes to central chemoreception, but the mechanisms are poorly understood[1, 2, 26]. Until now, in chemosensitive areas glutamate has been thought to act on neurons, and no mechanism involving astrocytes have been considered.

Here we report that mouse caudal medullary brainstem astrocytes contain D-serine and D-serine racemase, the enzyme that synthesizes D-serine from L-serine. We show that D-serine via activation of NMDAR increases the fR and, in awake unrestrained mice and brainstem slices, D-serine exerts its actions in specific brainstem nuclei like the raphe nucleus and the ventral respiratory column. By contrast, the reduction of D-serine levels, due to the inhibition of D-serine synthesis or the enzymatic degradation of D-serine, or the metabolic inhibition of astrocytes by sodium fluoroacetate decrease both the basal fR and the respiratory response to hypercapnia. Such depressed respiratory response induced by sodium fluoroacetate can be rescued by exogenous D-serine application. Our results reveal that D-serine is released by caudal medullary astrocytes and exerts a respiratory tonic drive and mediates the hypercapnia-induced respiratory response.

## Results

**Medullary astrocytes release D-serine during hypercapnia.** Immunofluorescence revealed that caudal medullary brainstem astrocytes contain D-serine (Fig. 1a, b) and express D-serine racemase (Fig. 1c, d) in primary tissue culture. The label for D-serine was uniformly distributed in the cytoplasm of the majority of the astrocytes as fine puncta (Fig. 1a), with a higher density around the nuclei. In contrast, the D-serine racemase was in dense spots distributed asymmetrically in the cytoplasm of astrocytes (Fig. 1c). After the culture medium was replaced with artificial cerebrospinal fluid (aCSF), medullary brainstem and cortical astrocytes were exposed to hypercapnic acidosis by switching the gas equilibration from 5 to 10% $CO_2$ in air. D-serine, L-serine, and L-glutamate were identified simultaneously

by liquid chromatography-tandem mass spectrometry (LC-MS/MS) from samples of the aCSF taken at different time points (Supplementary Fig. 1). Medullary brainstem astrocytes released D-serine and L-glutamate, but not L-serine, in response to hypercapnic acidosis (Fig. 1e–g). Compared with basal levels, the D-serine concentration in the extracellular medium increased by approximately three-fold after 5 min and six-fold after 30 min of hypercapnia (Fig. 1e, left). Thirty minutes after the restitution of basal conditions (5% $CO_2$), the D-serine level declined to that prior to hypercapnic acidosis (Fig. 1e, left). The glutamate concentration in the aCSF followed a similar time course with similar increases to those of D-serine during hypercapnic acidosis (Fig. 1g, left). After 5 and 30 min of hypercapnic acidosis, the glutamate concentration increased by two- and three-fold compared with the basal values, respectively, returning to baseline after 30 min of recovery in 5% $CO_2$. By contrast, no changes in the levels of D-serine, L-serine, or glutamate were observed during hypercapnic acidosis of cultures of neocortical astrocytes (Fig. 1e–g). At the end of the experiment, a 15 min stimulation with 50 mM KCl evoked a large, ~30-fold increase in extracellular D-serine and glutamate, but not L-serine, in medullary and in cortical astrocyte cultures (Fig. 1e–g). Thus, no change in the concentration of L-serine was detected during hypercapnic acidosis or KCl administration in medullary brainstem and cortical astrocytes (Fig. 1f). Only a very small increase in L-serine (Fig. 1f) was observed in cortical astrocytes during recovery after hypercapnia. This increase suggests a time-dependent cumulative effect. These results reveal that astrocytes in the medullary brainstem, but not in cortical regions, are able to respond to hypercapnia releasing D-serine and glutamate. In addition, these findings suggest that the D-serine and L-glutamate release depends on astrocyte depolarization and that D-serine and glutamate, likely, are co-released in response to hypercapnia.

To evaluate the contribution of extracellular calcium, gap junction hemichannels or vesicular mechanisms on hypercapnia-induced D-serine release, medullary brainstem astrocytes were incubated with calcium-free aCSF (aCSF containing 0 mM $Ca^{2+}$, 2 mM $Mg^{2+}$, and 1 mM EGTA) or with aCSF containing different pharmacological agents for 75 min under normocapnia (5% $CO_2$ equilibrated in air) followed by 30 min of hypercapnia (10% $CO_2$ equilibrated in air). The pharmacological agents were: 10 μM carbenoxolone or 1 mM probenecid to block pannexin-1 hemichannel, 100 μM carbenoxolone to block both connexin and pannexin-1 hemichannels, 2 μM bafilomycin A1 to inhibit vacuolar $H^+$ ATPase (V-ATPase) that provides the proton gradient necessary for intravesicular loading of gliotransmitters, and 50 μM brefeldin A to inhibit vesicle formation and transport between endoplasmic reticulum and the Golgi apparatus. Incubation with calcium-free aCSF elevated D-serine baseline from $281.2 \pm 127.6$ ($n = 7$) to $1355.0 \pm 306.3$ pmol per mg protein ($n = 6$) ($P = 0.0082$, Mann–Whitney test), suggestive of gate opening of connexin and pannexin-1 in calcium-free medium[27, 28]. Carbenoxolone (10 and 100 μM) or probenecid (1 mM) abolished the hypercapnia-induced release of D-serine, whereas it persisted, although reduced, in calcium-free aCSF (Fig. 1h). Similar results were obtained with high potassium-evoked D-serine release, which is also suggestive of pannexin-1 involvement (Fig. 1h). The hypercapnia- and high potassium-induced release of D-serine was also reduced with 2 μM bafilomycin A1, and to a lesser extent with 50 μM brefeldin A (Fig. 1h), suggesting the involvement of a vesicle release mechanism (Fig. 1h).

**D-serine modulates the respiratory rhythm.** The functional role of D-serine as a modulator of the respiratory rhythm was

evaluated in vitro and in vivo. Superfusion of aCSF containing D-serine (0.001–100 µM) increased the fR in both medullary brainstem slices and isolated brainstem-spinal cord en bloc preparations in a concentration-dependent manner (Fig. 2a, b; Supplementary Fig. 2). The maximal responses in medullary brainstem slices and en bloc preparations were $150.2 \pm 7.2\%$ ($n = 8$) and $156.3 \pm 5.9\%$ ($n = 13$) of the basal fR, with half maximal effective concentrations ($EC_{50}$) of $2.7 \pm 2.0$ and $1.9 \pm 1.8\,\mu M$, respectively (Fig. 2b; Supplementary Fig. 2a); these $EC_{50}$ values are within the range of the D-serine extracellular levels detected in the brain by microdialysis[29]. The amplitude of the integrated inspiratory burst did not change after the application of D-serine in medullary brainstem slices and en bloc preparations (Fig. 2c; Supplementary Fig. 2b). In addition, superfusion with aCSF containing 30 µM L-serine did not modify the fR that reached $102.9 \pm 1.9\%$ and $99.7 \pm 1.1\%$ of the basal fR in medullary brainstem slices ($n = 4$, $P = 0.2227$, Student's $t$-test) and en bloc preparations ($n = 4$, $P = 0.2901$, Student's $t$-test), respectively. The respiratory effects induced by adding 10 µM D-serine were suppressed by 10 µM dizocilpine (MK-801), an NMDAR antagonist (Fig. 2d), suggesting that D-serine exerts effects on respiratory neural networks by acting on functional NMDARs. A reduction in D-serine levels in medullary brainstem slices by exposure either to D-amino-acid oxidase (DAAO), an enzyme that degrades dextro amino acids, or inhibitors of D-serine racemase, such as phenazine methosulfate (MET-phen) and phenazine ethosulfate (ET-phen), reduced the basal fR (Fig. 2e). These results reveal the existence of a tonic respiratory drive depending on the endogenous release of D-serine in medullary brainstem slices.

The presence of D-serine-sensitive sites in regions of interest at the caudal surface of medullary brainstem slices (Fig. 3a) was determined by pressure-injecting (10 s, 3 psi) 9 nl of 300 µM D-serine through a glass micropipette (tip 3.75 µm ID). In the raphe nucleus (RN) and the ventral respiratory column (VRC), topical application of D-serine, but not vehicle, increased the frequency of the respiratory rhythm (Fig. 3b, c). Despite the

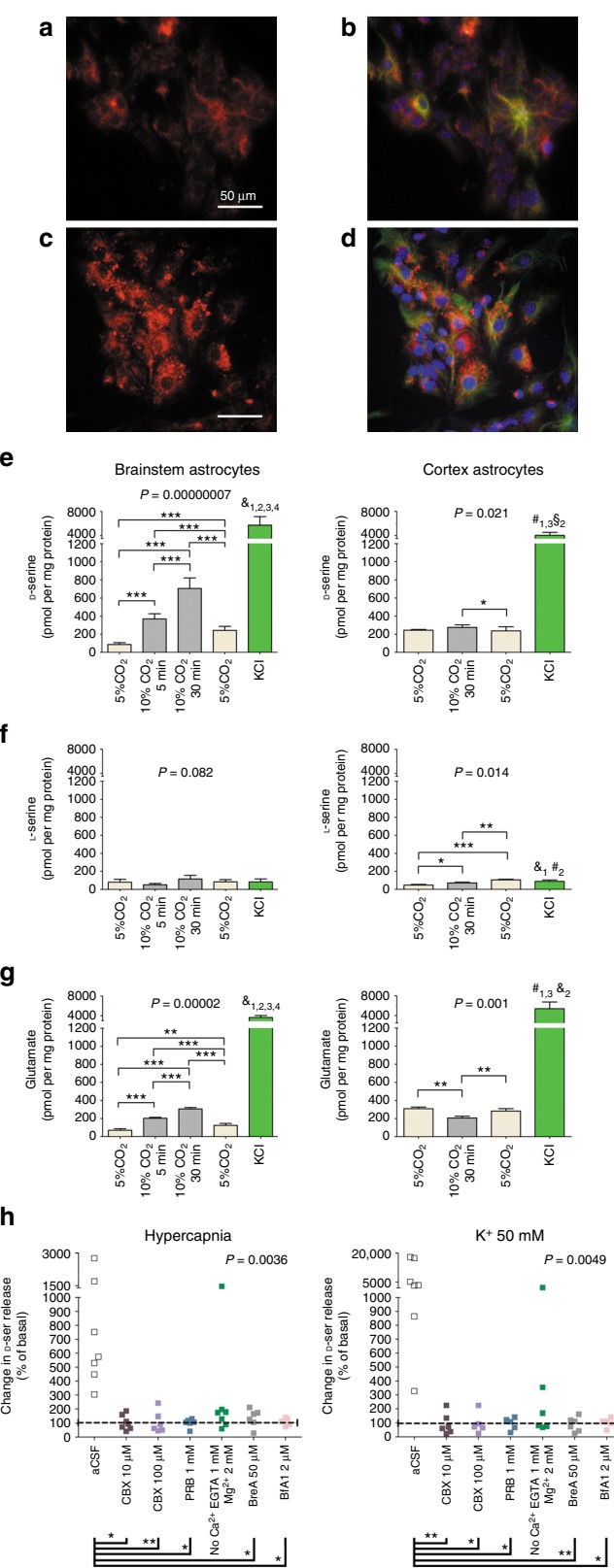

**Fig. 1** Medullary brainstem astrocytes release D-serine in response to hypercapnia. **a**, **b** (Red) D-serine and **c**, **d** (red) D-serine racemase were immunodetected in caudal medullary brainstem astrocytes. Merged microphotograph of D-serine and D-serine racemase in GFAP-positive astrocytes (green; **b**, **d**) and Hoechst (blue; **b**, **d**). **e–g** D-serine, L-serine, and glutamate were detected by LC-MS/MS in samples of aCSF taken from caudal medullary brainstem ($n = 5$) or neocortex ($n = 3$) astrocyte cultures at different time points. Measurements were performed after 30 min under basal conditions (5% $CO_2$), after 5 and 30 min of hypercapnic acidosis (aCSF equilibrated in 10% instead of 5% $CO_2$), and after 30 min of recovery (culture in a 5% $CO_2$ environment). At the end of the experiments, cultures were exposed to 50 mM KCl under basal conditions (5% $CO_2$) for 15 min. $P$ values obtained with Friedman's test are indicated for each group; multiple comparisons were performed with Conover's post hoc test; *, **, and *** indicate $P < 0.05$, $P < 0.01$, and $P < 0.001$, respectively; §, #, and &, indicate $P < 0.05$, $P < 0.01$, and $P < 0.001$, respectively for the KCl-evoked increase compared with the experimental conditions represented by each bar, where bar 1 represents the basal conditions. **h** Effects of hypercapnia (left) and high potassium (right) on D-serine concentration in the incubation medium of medullary astrocytes. Astrocytes were incubated in basal aCSF ($n = 7$) or calcium-free aCSF (CaCl$_2$ was replaced with MgCl$_2$ in aCSF containing 1 mM EGTA; $n = 7$) or aCSF containing 10 µM ($n = 7$) or 100 µM ($n = 6$) carbenoxolone or 1 mM probenecid ($n = 6$) or 2 µM bafilomycin A1 ($n = 5$) or 50 µM brefeldin A ($n = 6$). Astrocytes were exposed to pharmacological agents for 75 min under normocapnia (5% $CO_2$ equilibrated in air) in basal potassium (3 mM KCl) followed by hypercapnia (10% $CO_2$ equilibrated in air) in basal potassium (4 mM KCl) or by normocapnia in 50 mM KCl for 30 min. D-serine levels attained during hypercapnia or high potassium are expressed as percentage of respective basal levels for each culture at the different conditions. $P$ obtained with Kruskal–Wallis test is indicated; * and ** indicate $P < 0.05$ and $P < 0.01$, respectively (Dunn's multiple comparison post hoc test) between basal aCSF and conditions joined by horizontal lines at the bottom of the figure

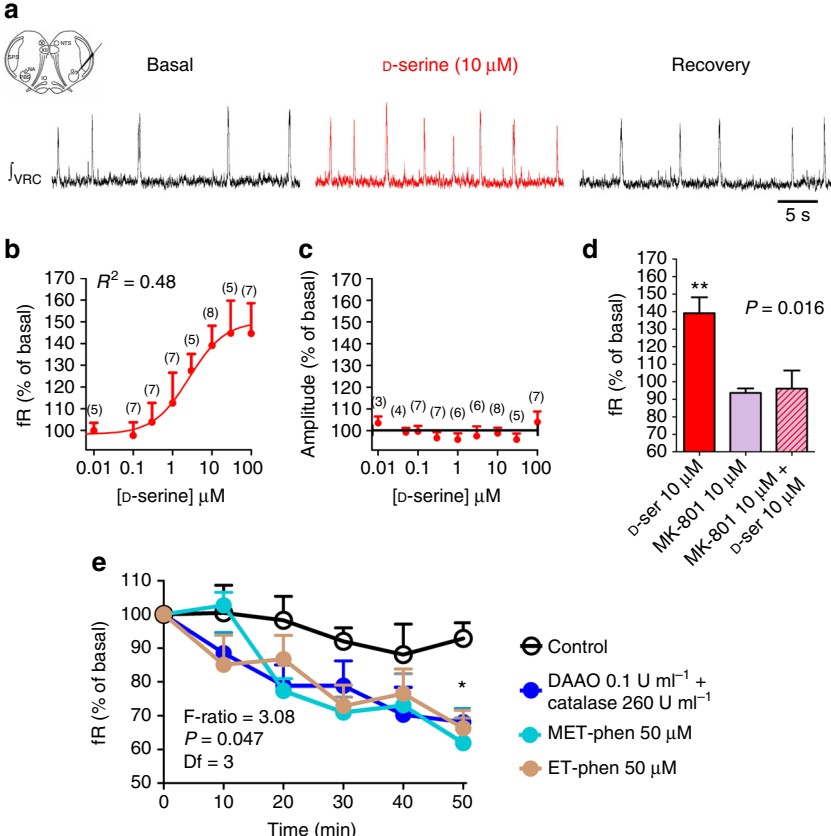

**Fig. 2** D-serine increases fR in medullary brainstem slices. **a** D-serine (10 μM) increased the fR recorded in slices from the ventral respiratory column (indicated in schematic). **b**, **c** Concentration–response curves for changes in frequency **b** and amplitude of the respiratory rhythm **c** induced by D-serine. The correlation coefficient ($R^2$) for the best fit to a sigmoidal curve and the number of slices for each concentration are shown. **d** The increase in fR induced by 10 μM D-serine (red bar) was abolished by 10 μM dizocilpine (MK-801, cross-hatched bar), whereas treatment with MK-801(purple bar) alone did not affect the basal fR ($n = 4$, $P = 0.0156$, Friedman's test; **$P < 0.01$, Conover post hoc test). **e** Superfusion for 50 min of medullary brainstem slices with aCSF containing DAAO ($n = 9$, blue circles), MET-phen ($n = 6$, cyan circles) or ET-phen ($n = 7$, brown circles) decreased the fR with respect to controls ($n = 5$, open circles; F-ratio = 3.08, $P = 0.047$ mixed ANOVA); * indicates $P < 0.05$, Bonferroni post hoc test. Symbols and vertical lines indicate the means and SEM, respectively

similarity in the duration and pressure of the pulses, the increase in frequency elicited by D-serine treatment in the RN underwent a different time-course, lasting longer than the frequency increase elicited at the VRC. These findings may indicate the triggering of additional local mechanisms or the existence of local differences in D-serine metabolism or accessibility. In contrast to the effects on the RN and VRC, the application of D-serine to the NTS, spinal trigeminal nucleus (5-SP), or intermediate regions did not evoke any respiratory effects (Fig. 3d, e).

Modulation of the respiratory rhythm by D-serine in adult, awake, and unrestrained CF1 mice was evaluated using whole-body plethysmography after either a single intraperitoneal injection (i.p.) or stereotaxic injection of D-serine (Fig. 4). Sixty minutes after an i.p. injection of 0.3 ml of saline solution containing D-serine to a final dose of 250 mg kg$^{-1}$, the minute volume ($V_E$) and fR, but not the tidal volume ($V_T$), increased significantly to $141.8 \pm 14.0\%$ ($V_E$), $124.8 \pm 5.4\%$ (fR), and $114.7 \pm 11.7\%$ ($V_T$) ($n = 6$, mean ± SEM, Fig. 4a–c). The time course for the increment in fR after a single i.p. injection of D-serine was similar to that reported for the increase in D-serine brain levels in rats[30]; fR reached a peak 60 min after an i.p. injection of D-serine, and decayed slowly over the next 240 min, returning to its basal value (Fig. 4c). By contrast, 60 min after an i.p. injection of 0.3 ml of L-serine in saline to a final dose of 250 mg kg$^{-1}$, $V_E$ was not modified and reached $105.0 \pm 3.9\%$ of the basal $V_E$ ($n = 4$, $P =$

0.2901, Student's t-test). In addition, i.p. injection of saline alone ($n = 6$) also failed to change any ventilatory parameters within the same time period (Fig. 4a–c). Although the existence of D-serine and D-serine racemase in carotid bodies has not been reported, they express NMDAR in glomus cells[31] that may be involved in D-serine-evoked response. To show unequivocally that D-serine can modulate the fR in adult awake unrestrained mice, D-serine was injected directly into the brainstem. Based on our findings in slices that showed that D-serine applied topically in the RN evoked an important increase in the fR (Fig. 3), stereotaxic cannula implantations were targeted into the RN in adult mice. Two minutes after a single injection of 0.2 μl aCSF containing 30–300 μM D-serine into the RN of awake, unrestrained mice, fR increased to a peak reaching to $142.7 \pm 12.5\%$ ($n = 6$) of the baseline (Fig. 4d, g, h). Then, within 2 min after the peak, fR decayed to baseline values (Fig. 4h). In contrast, 0.2 μl of aCSF alone did not evoke any change in fR (Fig. 4f, h).

**D-serine mediates the respiratory response to hypercapnia.** The role of D-serine as a mediator of the respiratory response to hypercapnia was also evaluated both in medullary brainstem slices and awake unrestrained mice. Medullary brainstem slices were challenged with hypercapnic acidosis when the superfusion was switched from aCSF gassed with 5% $CO_2$ (pH 7.4) to aCSF gassed with 10% $CO_2$ (pH 7.2) equilibrated in oxygen.

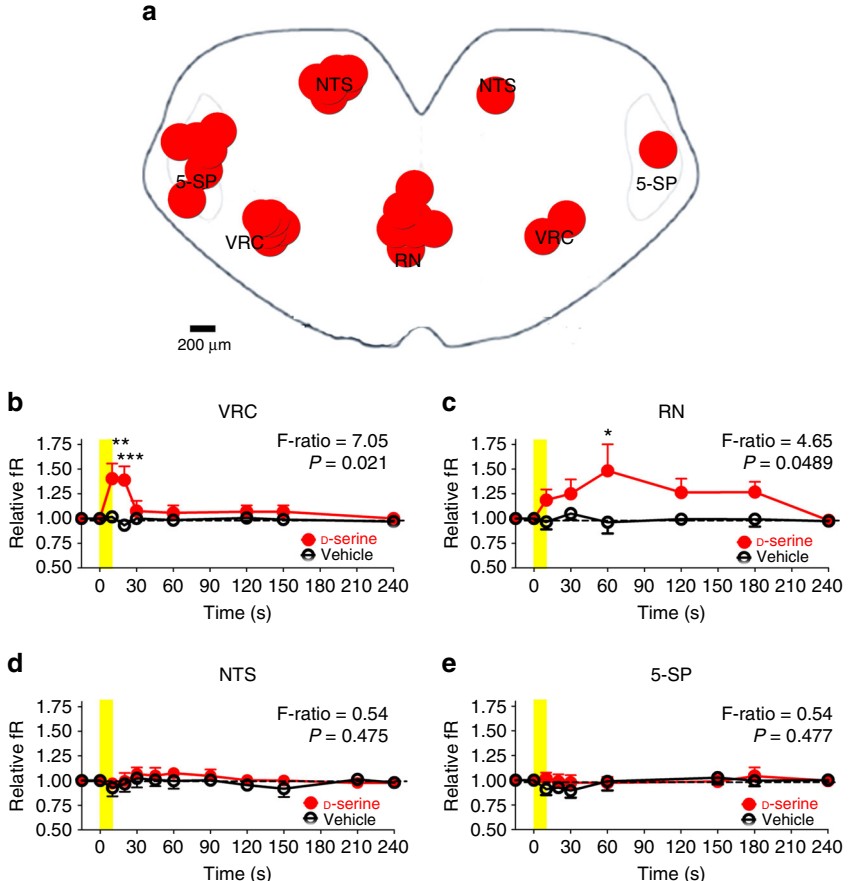

**Fig. 3** Focal application reveals D-serine sensitive nuclei. **a** Schematic of a medullary brainstem slice illustrating the sites of injection of D-serine and the extension of the labeled area when a bolus of aCSF containing 1% neutral red dye was pressure-ejected onto a specific locus. **b–e** Time course of changes in relative fR (ratio between fR at a given time and basal fR) induced by 10 s, 3 psi injections of 300 µM D-serine (red filled circles) and vehicle (aCSF pH 7.4, open circles) in specific brainstem nuclei. Injections of D-serine into the VRC (**b**) or the RN (**c**) increased the fR in neonatal medullary brainstem slices, but not when applied to the NTS (**d**) or the 5-SP (**e**). The yellow vertical bar indicates when a 10 s pulse was applied. Horizontal dashed line indicates no change. Symbols and vertical lines, mean and SEM, respectively ($n = 7$ for VRC, RN and NTS; $n = 6$ for 5-SP). The mixed ANOVA F-ratio and P are indicated for each nucleus; *$P < 0.05$, **$P < 0.01$, ***$P < 0.001$ Bonferroni post hoc test. 5-SP spinal trigeminal nucleus, NTS nucleus tractus solitarius, RN raphe nucleus, RTN retrotrapezoid nucleus, VRC ventral respiratory column

Hypercapnic acidosis increased the fR to $157.6 \pm 8.2\%$ ($n = 33$) with respect to the baseline values in slices from control mice (Fig. 5a). The elimination of D-serine after the exposure of medullary slices to DAAO for 45 min significantly reduced the $CO_2$-induced respiratory response (Fig. 5b, c). In addition, the impairment of D-serine synthesis via exposure of the slices to MET-phen or ET-phen for 45 min abolished the respiratory response to hypercapnic acidosis (Fig. 5d–g). Similar results were obtained in en bloc preparations (Supplementary Fig. 4a–d). The $CO_2$-induced respiratory response was also abolished by exposing medullary slices to MK-801, suggesting that the respiratory response relies on a NMDAR-related mechanism (Fig. 5h, i). Deterioration of the slice preparation during the period in which the experiments were performed was not observed. In fact, hypercapnic acidosis increased the fR similarly at the beginning and the end of the 45–60 min superfusion with basal aCSF in both medullary brainstem slices and en bloc preparations (Fig. 5j, k; Supplementary Fig. 4e, f). Thus, the loss of responsiveness to $CO_2$ could not be attributed to a reduction in basal fR induced by DAAO. Spearman correlation analysis revealed that the largest increases in respiratory rate induced by $CO_2$ in control and DAAO-treated slices were in those slices that had the lowest basal respiratory rates (Supplementary Fig. 5).

Adult mice received i.p. injections with 0.3 ml of saline (controls) or saline containing 9 mg kg$^{-1}$ MET-phen (to inhibit D-serine racemase), and ventilatory responses to hypercapnia were evaluated by whole-body plethysmography. Four hours after the MET-phen injection, the basal fR was significantly reduced (Supplementary Fig. 3b). The hypercapnic test was performed in awake unrestrained mice breathing room air for 5 min (basal) followed by air containing 10% $CO_2$ for 10 min (hypercapnia) and then back to room air for 5 min (recovery). In control mice, hypercapnia was associated with increased minute volume of $206.7 \pm 9.3\%$ of basal value, which depended on increases in fR and $V_T$ of $129.5 \pm 8.1\%$ and $159.8 \pm 7.1\%$ ($n = 19$) of the basal values, respectively (Fig. 5l). In mice in the MET-phen group, hypercapnia was also associated with increased minute volume, fR and $V_T$. The magnitude of the changes in $V_T$ was similar among mice in the control ($n = 7$) and MET-phen ($n = 12$) groups. $V_T$ during hypercapnia in the saline-injected group reached $169.4 \pm 17.7\%$, $159.0 \pm 15.5\%$, $146.5 \pm 6.9\%$, and $164.5 \pm 10.4\%$ of baseline at 0, 2, 4, and 24 h after i.p. saline injection. Whereas, $V_T$ during hypercapnia in the MET-phen-injected group reached $154.1 \pm 4.6\%$, $144.0 \pm 6.2\%$, $153.8 \pm 6.7\%$, and $154.3 \pm 5.2\%$ of baseline at 0, 2, 4, and 24 h after i.p. MET-phen injection ($P = 0.2368$, F-ratio $= 1.504$, df $= 1$, mixed ANOVA). However, the magnitude of fR changes induced by hypercapnia

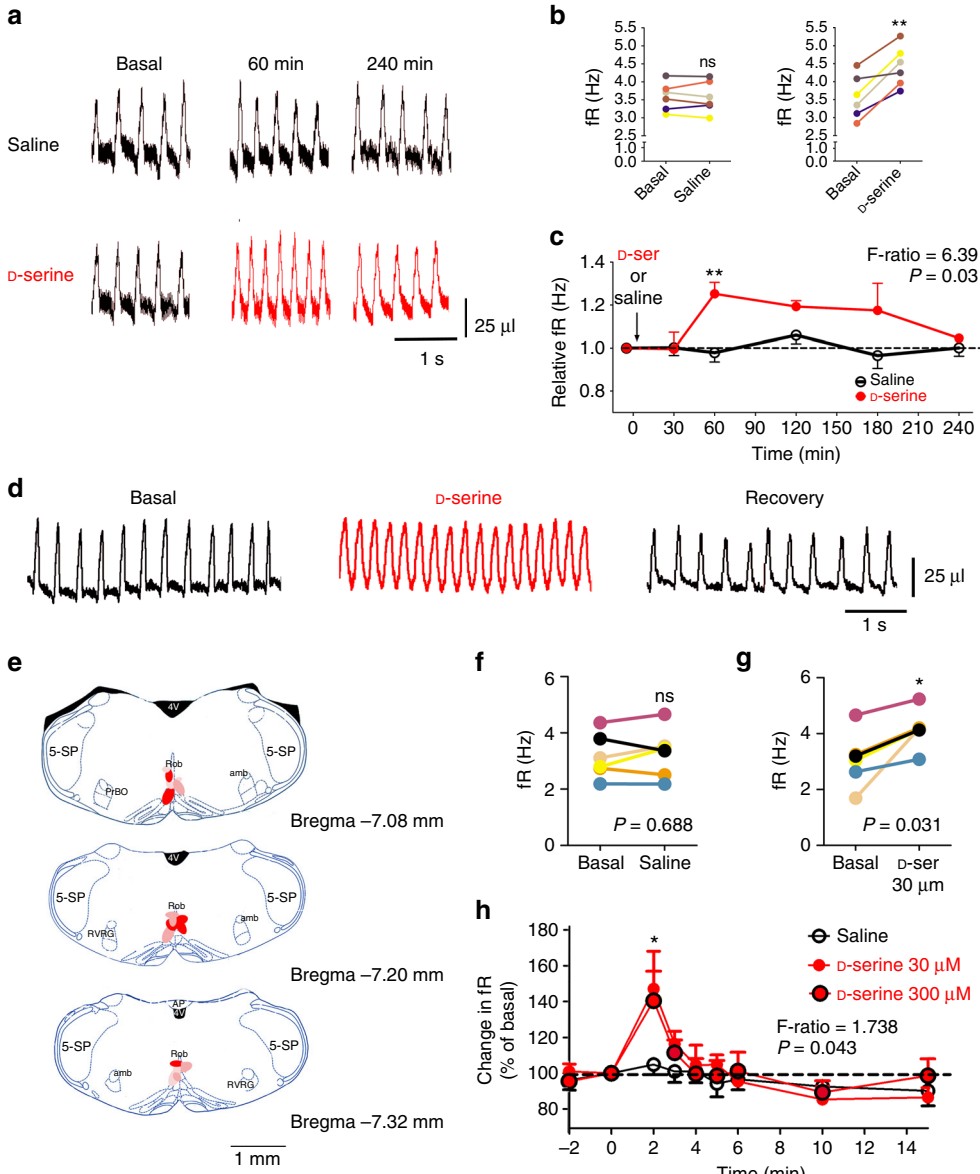

**Fig. 4** D-serine increases fR in awake unrestrained mice. **a** Ventilatory recording in an adult mouse spontaneously breathing air before (basal) and 60, and 240 min after an i.p. injection of saline (control) or 250 mg kg$^{-1}$ D-serine. **b** fR before (basal) and 60 min after an injection of saline (left, $n = 6$) or D-serine (right, $n = 6$) for each mouse, represented by colored lines; ns not significant; **, $P < 0.01$ (Wilcoxon signed-rank test). **c** Relative fR (ratio between fR at a given time and basal fR) after a single injection of D-serine (F-ratio = 6.39, $P = 0.03$, mixed ANOVA); **, $P < 0.01$ Bonferroni post hoc test. **d** Ventilatory recordings in awake unrestrained mice breathing air before (basal), after 2 min, and after 12 min of 0.2 µl injection of aCSF containing 30 µM D-serine through a cannula implanted and targeted into the RN, under isofluorane anesthesia 4 days before the experiments. **e** Diagram of the localization of the injection sites into the RN (red spots) visualized with methylene blue dye injected at the end of the experiments. **f**, **g** fR of six adult mice before (basal), and 2 min after injection of saline (**f**) or 30 µM D-serine (**g**) into the RN. Colored lines connect the fR values before and 2 min after injections for each mouse. ns not significant; * indicates $P < 0.05$, Wilcoxon signed-rank test. **h** Time course of the fR in six adult mice before (basal) and after injection of saline (black open symbols and lines) or 30 or 300 µM D-serine (red filled symbols and lines). Time zero corresponds to the injection time. fR is expressed as percentage of basal values (before injection). Dashed horizontal line indicates no change in fR. The mixed ANOVA F-ratio and P are indicated; *$P < 0.05$, Bonferroni post hoc test

were lower in the mice in the MET-phen group than in control mice at 2 and 4 h after injection (Fig. 5m). These fR changes led to a significant reduction of the minute volume in response to hypercapnia in mice of the MET-phen group ($n = 12$) compared with saline-treated mice ($n = 6$) 2 h after injection (F-ratio = 12.14, $P = 0.003$, mixed ANOVA $P < 0.05$, Bonferroni post hoc test). The hypercapnia-induced changes in fR were not correlated with the basal fR (Supplementary Fig. 3c, d). Spearman correlation analysis indicated a significant negative correlation between the hypercapnia-induced increase in fR and the basal fR

in control mice, whereas in MET-phen-injected mice the two variables were uncorrelated (Supplementary Fig. 3c, d). The effect of the MET-phen injection was reversible, and 24 h later, both the basal fR and hypercapnia-induced fR changes returned to the baseline values (Fig. 5m).

**D-serine restores the deficit due to metabolic inhibition of astrocytes.** The respiratory rhythm in slices of neonatal mouse is depressed by sodium fluoroacetate[32], a selective metabolic

inhibitor of the Krebs cycle in astrocytes[32]. Addition of glutamine, known to be reduced by fluoroacetate, and the main precursor for glutamate synthesis in the central nervous system, restored and even enhanced the respiratory rhythm[32]. Thus, superfusion of brainstem slices with fluoroacetate in the presence

of glutamine depresses astrocytes but preserves the respiratory network activity. To link astrocyte function with D-serine release, hypercapnic test was performed in medullary brainstem slices superfused initially with aCSF supplemented with 1.5 mM glutamine for 30 min (basal condition) followed by superfusion with

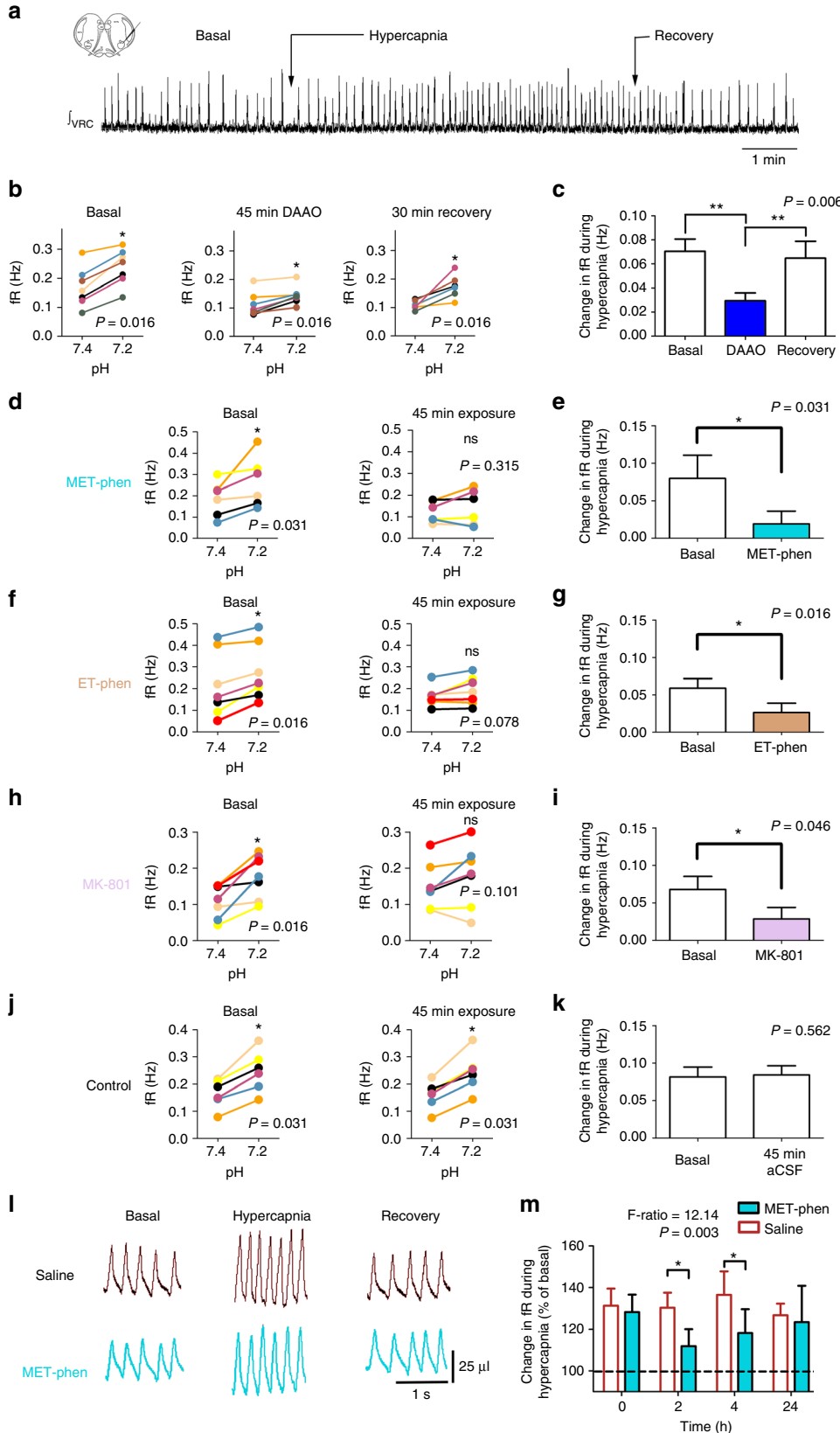

glutamine-supplemented aCSF containing 5 mM fluoroacetate for additional 30 min. We confirmed that superfusion with glutamine, before fluoroacetate administration, increased the fR to 130 ± 1% of basal ($n = 7$, $P < 0.05$, Wilcoxon signed-rank test)[32]. We found that the hypercapnia-induced respiratory response was similar to that observed in slices superfused with aCSF lacking glutamine supplementation, and this response was preserved after several hours of glutamine administration (Fig. 6a). In contrast, addition of fluoroacetate to the glutamine supplemented aCSF for 30 min impaired the respiratory response to hypercapnia (Fig. 6b). Notoriously, 50 μM D-serine treatment succeeded in restoring the hypercapnia-induced respiratory response (Fig. 6c). Furthermore, in a separate set of experiments, we evaluated whether fluoroacetate affected the $CO_2$-evoked increase in D-serine levels in a 1.5 ml incubation medium containing three caudal brainstem slices. Hypercapnic acidosis was performed after 30 min of incubation with glutamine-supplemented aCSF in normocapnic conditions (gassed with 5% $CO_2$ equilibrated in $O_2$) by switching the gassing to 10% $CO_2$ equilibrated in $O_2$ for 10 min. After 20 min of recovery, 10 min of hypercapnic acidosis was repeated at the end of a period of incubation with glutamine-supplemented aCSF containing sodium fluoroacetate 5 mM for 30 min. We found that fluoroacetate treatment reduced the increase in D-serine content in the incubation medium induced by hypercapnic acidosis (Fig. 6e). This result agrees with the fluoroacetate reduction in the respiratory response to hypercapnia and suggests that astrocytes are the source of D-serine during hypercapnic acidosis.

## Discussion

The work presented here shows that D-serine is a component of the interoceptive mechanisms that generate the central respiratory response to $CO_2$, establishing a role for D-serine in addition to the roles of D-serine in synaptic plasticity, cellular migration, cell death, neurotoxicity, and neurodegeneration[33]. We show that mouse caudal medullary brainstem astrocytes in culture can synthesize, store, and release D-serine and glutamate in response to hypercapnia. In contrast, neocortical astrocytes, which are able to sense $CO_2$ increasing their intracellular calcium concentration[5], were unresponsive in terms of D-serine and glutamate release. Whether neocortical astrocytes use similar mechanisms to those described in rostral brainstem astrocytes[34] to sense $CO_2$ is an open question. Our results indicate that medullary brainstem astrocytes differ from cortical astrocytes, at least, in the specific cellular response triggered by chemosensitive stimulation. D-serine, through activation of NMDAR, activated the respiratory network, driving the timing of the respiratory rhythm. Respiratory effects were observed both in awake, unrestrained adult mice, and in caudal medullary brainstem slices from mouse neonates. Focal injection of D-serine into brainstem nuclei in slices

identified sensitive sites in the caudal medullary brainstem, like the RN and the VRC. Stereotaxic injection of D-serine into the RN in awake, unrestrained adult mice confirmed that a D-serine mechanism can operate in this nucleus in physiological conditions. Inhibition of D-serine synthesis or enzymatic degradation of D-serine blunted the hypercapnia-induced increase in fR and reduced the basal fR, revealing a D-serinergic tonic respiratory drive. Furthermore, exogenous D-serine was enough to restore the hypercapnia-induced respiratory response after it was depressed in brainstem slices by the selective metabolic inhibition of astrocytes. Thus, the hypercapnia-induced release of D-serine and L-glutamate by caudal medullary brainstem astrocytes can account for the glutamatergic contribution to the central chemoreception[26].

Preliminary exploration of possible mechanisms responsible of the $CO_2$-evoked D-serine release suggests that gap junction hemichannels, likely pannexin1, and vesicle release mechanisms are involved. The incubation with aCSF without calcium, a condition known to trigger opening of gap junction hemichannels[27, 28, 35] increased the D-serine basal level. The $CO_2$-evoked D-serine release was reduced by probenecid and by 10% carbenoxolone, both selective blockers of pannexin-1 hemichannel and ineffective blockers of connexins[36]. In addition, both pannexin-1 blockers also inhibited the high potassium-induced release of D-serine, which also supports the involvement of pannexin-1, because pannexin-1 is activated by high extracellular potassium and membrane depolarization[37, 38]. Pannexin-1 is expressed in several cells. In particular, in astrocytes[39–42], where it plays a role as a major ATP release channel[37, 38, 42]. In addition, pannexin-1 forms a complex with the purinergic P2X7 receptor (P2X7R-pannexin-1 complex) that seems to be crucial for P2X7R-mediated D-serine release[43]. By contrast, $CO_2$-sensitive astrocytes in the rostral brainstem release ATP in response to hypercapnia through connexin hemichannels and not through pannexin-1[35]. Impairment of processes related to exocytosis affected the release of D-serine from medullary astrocytes. The effects of bafilomycin A1 and brefeldin A suggest that the vesicular proton gradient and the formation of vesicles have a role in the mechanisms of D-serine storage/release. It is worth to note that exocytosis of other gliotransmitter, for instance glutamate, may be affected and this could drive D-serine release from medullary astrocytes as has been described for cortical astrocytes[44]. Then, the impairment of vesicular mechanisms, indirectly, could affect D-serine release by non-exocytotic conduits.

Because synaptic NMDARs are preferentially gated by endogenous D-serine[22], D-serine released in response to hypercapnia could preferentially modify glutamatergic synaptic efficacy (Fig. 7). In addition, D-serine and L-glutamate released by astrocytes at extrasynaptic sites can also alter neuronal and network excitability via extrasynaptic NMDARs[45]. It is likely that

**Fig. 5** D-serine level reduction or NMDAR blockade decreases the respiratory response to hypercapnia in medullary brainstem slices and adult mice. **a** Integrated inspiratory burst recorded from the VRC in slices obtained from P4 neonates during isohydric normocapnia (basal, 5% $CO_2$, pH 7.4), hypercapnic acidosis (10% $CO_2$, pH 7.2), and recovery to basal conditions. **b, d, f, h** Exposure to 0.1 U ml$^{-1}$ DAAO + 260 U ml$^{-1}$ catalase (**b**, $n = 7$) to degrade D-serine, exposure to 50 μM MET-phen (**d**, $n = 6$) or 50 μM ET-phen (**f**, $n = 7$) to inhibit D-serine racemase, and exposure to 10 μM MK-801 (**h**, $n = 7$) for NMDAR blockade for 45 min each reduced the hypercapnia-induced fR response in medullary brainstem slices. Colored lines connect the fR values at pH 7.4 and 7.2 for each medullary brainstem slice before exposure (basal, first column of graphs), after 45 min of exposure (second column of graphs), and after 30 min of wash out (**b**, recovery). **c, e, g, i** Average changes in fR induced by hypercapnia under control conditions (basal, white bars); after 45 min of superfusion with aCSF containing DAAO (**c**, $n = 7$, blue bar), MET-phen (**e**, $n = 6$, cyan bar), ET-phen (**g**, $n = 7$, brown bar), or MK-801 (**i**, $n = 7$, purple bar); and after 30 min of recovery (second white bar in **c**). By contrast, 45 min superfusion with basal aCSF (slice responsiveness control) did not modify the fR (**j**, $n = 6$) or average responses (**k**, $n = 6$). Average data are presented as the mean + SEM (bars and vertical lines, respectively). Analysis for **c**: $P = 0.0062$, Friedman's test; **, $P < 0.01$ Conover post hoc test. Analysis for **b, d–k**: ns not significant; * indicates $P < 0.05$, Wilcoxon signed-rank test. **l** Ventilatory response to hypercapnia in adult mice injected with saline (control) or saline with 9 mg kg$^{-1}$ MET-phen. **m** Average fR response to hypercapnia expressed as the percentage of the basal fR in mice injected with saline ($n = 7$, controls) and MET-phen ($n = 12$) at different time points after injection ($P = 0.017$, mixed ANOVA; *, $P < 0.05$ Bonferroni post hoc test)

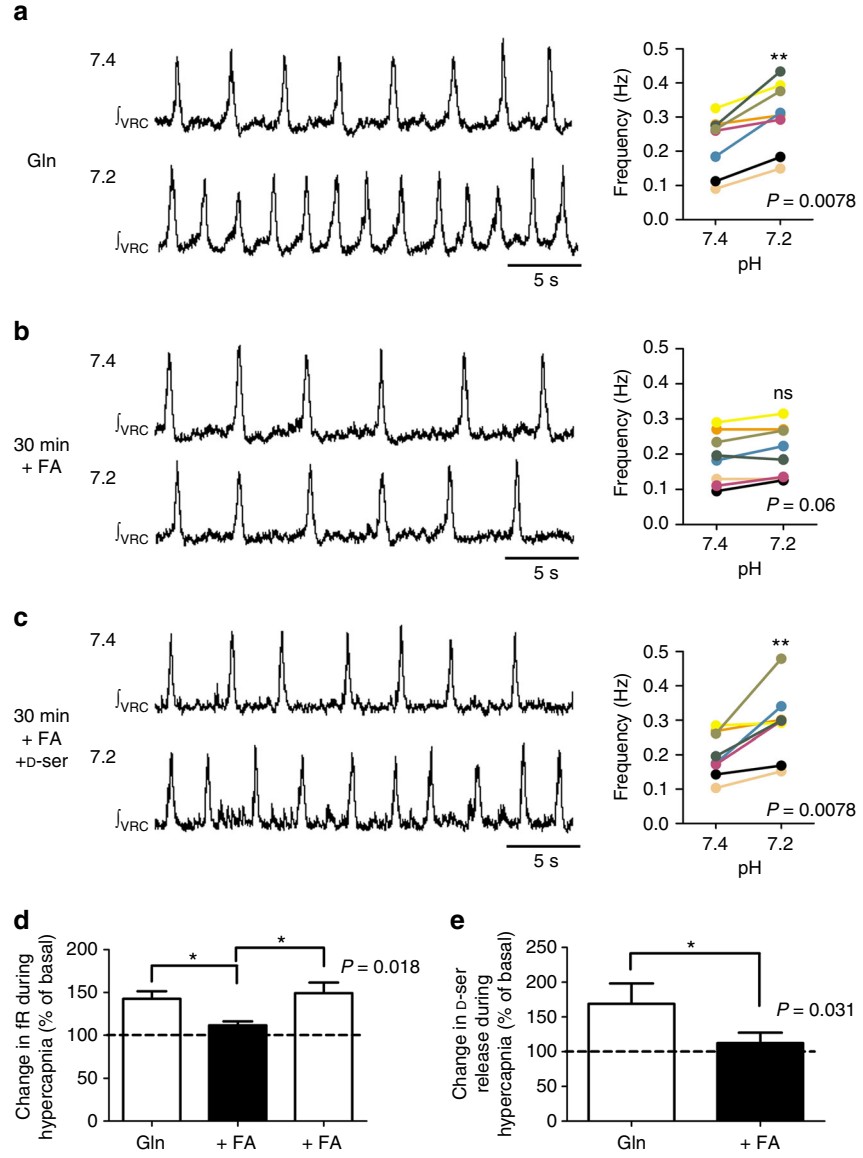

**Fig. 6** Exogenous D-serine restores hypercapnia-induced respiratory response after depression by metabolic inhibition of astrocytes. Integrated inspiratory burst recorded from VRC in eight brainstem slices obtained from P4 neonates and superfused with glutamine (Gln)-supplemented aCSF for 30 min (**a**, left) then with Gln-supplemented aCSF containing 5 mM fluoroacetate (FA) for 30 min (**b**, left) and finally with Gln-supplemented aCSF + 5 mM FA + 50 μM D-serine (**c**, left) during isohydric normocapnia (upper traces, 5% $CO_2$, pH 7.4) and hypercapnic acidosis (lower traces, 10% $CO_2$, pH 7.2). To the right of recordings, colored lines connect the fR values at pH 7.4 and 7.2 for each medullary brainstem slice ($n = 8$) after 30 min of isohydric normocapnia (pH 7.4) and hypercapnic acidosis (pH 7.2) for 10 min; ** indicates $P < 0.01$, Wilcoxon signed-rank test; ns not significant. **d** Average changes in fR induced by hypercapnic acidosis, expressed as percentage of fR values in isohydric normocapnia after 30 min of superfusion with Gln-supplemented aCSF, Gln-supplemented aCSF + FA, and Gln-supplemented aCSF + FA + D-serine ($n = 8$, $P = 0.018$, Friedman's test); multiple comparisons were performed with Conover's post hoc test; * indicates $P < 0.05$. **e** The hypercapnia-induced increment of D-serine levels in 1.5 ml incubation medium containing three caudal brainstem slices was abolished by fluoroacetate (FA). * indicates $P = 0.031$ ($n = 7$, Wilcoxon signed-rank test). Changes during hypercapnic acidosis are expressed as percentage of D-serine levels in isohydric normocapnia

astrocytic release of ATP in the RTN and of D-serine in the caudal medullary brainstem (RN and VRC) amplify and intensify the excitation of the respiratory network by intrinsic $CO_2$/$H^+$-sensitive neurons (Fig. 7).

Conditional deletion of D-serine racemase (SR) in astrocytes, using a tamoxifen-inducible GFAP + astrocyte-specific Cre-recombinase-mediated SR knockout[46], reduced the expression of D-serine racemase by only 15% in the mouse forebrain, the brain region with the highest D-serine racemase levels[46]. Such small decrease in D-serine racemase did not affect the D-serine

levels nor the hipoccampal LTP induction[46]. In contrast, the conditional deletion of neuronal D-serine racemase resulted in a 60% reduction of the enzyme in the cerebral cortex and hippocampus and was associated with hippocampal LTP impairment[46]. In general, neurons exhibit relatively high expression of serine racemase[47, 48] and synthesize more D-serine than astrocytes[46]. However, neurons has a smaller amount of D-serine than astrocytes[47, 49], with the exception of brainstem and cerebellar astrocytes, which exhibit a relatively low level of D-serine, likely due to the high DAAO content[50]. Astrocytes, but not neurons are able to

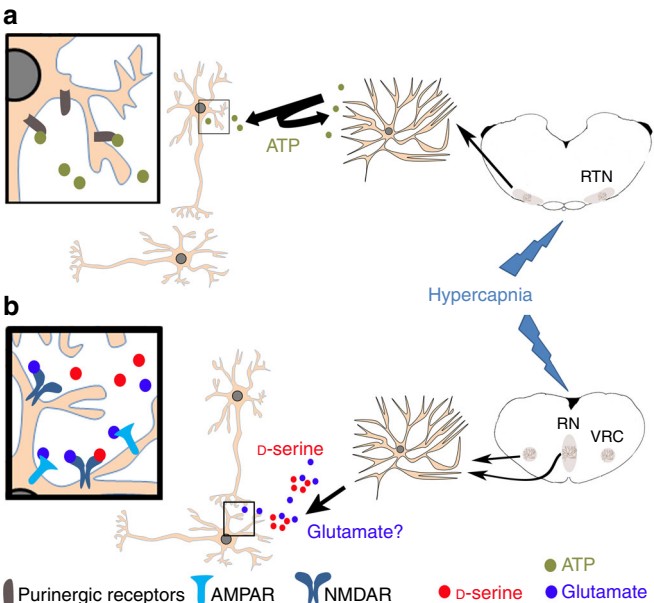

**Fig. 7** D-serine released by astrocytes is a mediator of central chemoreception. **a** The ATP released by astrocytes in response to hypercapnia acts on the purinergic receptors of RTN neurons. The inset shows purinergic receptors in dendrites and the cell soma. **b** D-serine and glutamate are released by astrocytes of the RN and the VRC in response to hypercapnia. The inset shows synaptic NMDARs. D-serine and glutamate that can also act on extrasynaptic NMDARs in the respiratory network. Thus, D-serine, as a glutamate co-agonist, can modify either the synaptic efficacy by acting on synaptic NMDARs, or the excitability of neurons by acting on extrasynaptic NMDARs. RN raphe nucleus, RTN retrotrapezoid nucleus, VRC ventral respiratory column

synthesize L-serine, the precursor for the D-serine synthesis. In addition, in contrast to neurons, astrocytes are able to store D-serine. Thus, a major fraction of the D-serine stored by astrocytes are synthesized by neurons[49]. In fact, the serine shuttle hypothesis proposes that astrocytes synthesize and provide L-serine to neurons that synthesize D-serine, which is released to the environment and taken up by astrocytes where is stored and released, for instance, in response to hypercapnia[49]. In addition to astrocytes, the caudal medullary brainstem also contains $CO_2$-sensitive neurons. For example, serotonergic RN neurons, which are sensitive to $CO_2$[51], may be a source of D-serine. Even if this was the case, astrocytes should be considered as key elements for central respiratory chemoreception in caudal medulla either because of D-serine release or their contributions to the serine shuttle as a source of L-serine for the neuronal synthesis of D-serine[49]. However, the metabolic inhibition of astrocytes, but not neurons by fluoroacetate reduced the hypercapnia-induced response in slices suggesting strongly that astrocytes are crucial for central chemoreception. Even more, the rescue of the chemosensory response, during metabolic inhibition of astrocytes, by exogenous D-serine agrees with the proposition that $CO_2$-sensitive astrocytes are the source of the D-serine that is released during hypercapnia.

Our data reveal that D-serine is a relevant factor that modulates the respiratory rhythm in both neonates and adult mice. Because results in neonates were obtained with in vitro experimental approaches and those in adult with experiments in awake unrestrained mice, any comparison between the contribution of D-serine to neonatal and adult central chemoreception should be cautious. Slice preparations obtained from neonates allow a reliable control of the superfusion medium and an easy access to specific components of the neural network. However, they lack

important afferents provided by peripheral organs, like the vagal mechanoreceptors and arterial peripheral chemoreceptors, as well as central chemosensory structures, like the RTN. Additionally, different experimental conditions (temperature, extracellular potassium concentration) can impact the network excitability, and hence influence the respiratory response to hypercapnia. At different postnatal ages, the relative contribution of different nuclei to the central chemoreception may vary. In the RN, serotonergic (5-HT) neurons require to mature during the first few weeks of life to contribute effectively to respiratory $CO_2$ and pH chemoreception in mice in vivo[52, 53]. Our data reveal that the mechanisms triggered by D-serine in the RN are already functional at the first week of postnatal life, being already able to modulate the respiratory rhythm.

In conclusion, our results demonstrate a role for D-serine as a mediator of the hypercapnia-induced respiratory response in caudal medullary chemosensory nuclei. In contrast to ATP, which excites respiratory neurons[4, 7], D-serine modulates glutamatergic synapse efficacy[33, 54, 55] in the respiratory network. We propose that D-serine released with glutamate from $CO_2$-sensitive astrocytes accounts for the glutamatergic contribution to central chemoreception.

## Methods

**Animals.** Experiments were performed in healthy non-medicated newborns at postnatal days 0–4 (P0-P4) and 3–4 months old (40–50 g) male and female CF1 mice. They were carried out in accordance with the Institute for Laboratory Animal Research (ILAR) Guide for the Care and Use of Laboratory Animals, with the approval of the Bioethics Committee of the Comisión Nacional de Investigación Científica y Tecnológica de Chile (CONICYT) and the Bioethics Committee of the Universidad de Santiago de Chile. In each experimental group, newborn mice were obtained at random from at least 3–5 different litters. Adult animals also were also chosen at random and no further exclusion criteria was used.

**Ventilatory recordings.** Ventilation was recorded in awake and freely moving adult CF1 mice using whole-body plethysmography under normoxic normocapnia or hypercapnia conditions. Mice were placed in a chamber with a volume of 900 ml and flow of 180 ml min$^{-1}$ and thermoregulated at 28–30 °C by a temperature controller (FHC Bowdoin, ME, USA). The controller commanded an electric mantle in close contact with the chamber while the temperature inside the apparatus was monitored with a thermistor (YSI, 44107). Plethysmographic recordings were performed after 30 min of habituation inside the chamber. Ventilation was recorded during the last 30 s of each exposure with the plethysmography chamber hermetically sealed. When the inflow stopcock was closed, the outflow stopcock was connected to one of the two ports of a differential pressure transducer (spirometer pod ML-311, ADInstruments, New South Wales, Australia), and the other port remained opened to the environment. The signal from the spirometer was digitized at 1 kHz with an A/D acquisition system PowerLab (4/25T model, ADInstruments, New South Wales, Australia) connected to a computer controlled by LabChart 5.0 software (ADInstruments, New South Wales, Australia) for on-line oscilloscope mode display and subsequent data analysis. Calibration was performed by the injection of 20, 50, and 100 μl through a small port into the recording chamber with a 20 or 100 μl Hamilton syringe (Hamilton company, Reno, NV). The mixture of gases was humidified, and the flow pressure maintained at 5 cm $H_2O$ by passing gases through the tip of a tube submerged 5 cm below the water surface in a bottle whose outflow was connected to the entrance of the chamber. The administration of pure air through this system did not evoke any consistent changes in the amplitude or frequency of respiratory rhythm.

**Hypercapnia challenge in vivo.** A typical recording under hypercapnia conditions consisted of an initial period of basal conditions in which mice breathed air (21% $O_2$ equilibrated in $N_2$) for 5 min, followed by a period of hypercapnia in which mice inhaled air enriched with 10% $CO_2$ for 10 min, and finally a period of recovery in which mice breathed air again for 5 min. Measurements of ventilatory variables were performed during quiet periods in which the recordings were free of movement artifacts and the mice breathed regularly for at least 30 s at the end of each exposure period.

**D-serine and serine racemase inhibition experiments in vivo.** D-serine (250 mg kg$^{-1}$) dissolved in saline ($n = 6$, D-serine) or saline alone ($n = 6$, control) was administered in a single intraperitoneal (i.p.) injection of 0.3 ml. In another set of experiments, a single injection of 0.3 ml of saline ($n = 6$, controls) or saline containing 9 mg kg$^{-1}$ MET-phen ($n = 12$) was administered.

MET-phen and ET-phen have been used previously as D-serine racemase inhibitors: (1) in neonatal mice, 3 mg kg$^{-1}$ per day MET-phen i.p. injections for 3 days reduced the D-serine levels 24 h later to 86.4% of the control levels in the prefrontal cortex and 81.2% in the cerebellum[56]; (2) in the tiger salamander retina in vitro, 10 µM ET-phen reduced the D-serine content by 50%, with no significant changes in the L-serine levels[57]; (3) in primary cultured mouse spinal cord cells, 1 µM MET-phen reduced the serine racemase (L-ser β-elimination) activity to 64.5%[58]; L-ser β-elimination is a secondary enzymatic reaction in which pyruvate is synthesized from L-serine. This reaction has a similar $K_m$ value and the same enzymatic active site as the reaction for the isomerization of L-serine to D-serine[59]. MET-phen and ET-phen have been used to inhibit granule cell migration along cerebellar Bergmann glia; the progression of this process depends on the D-serine levels[60]. MET-phen and ET-phen demonstrated an $IC_{50}$ of 3 and 5 µM, respectively, in the reduction of granule cell migration; the reduction induced by 50 µM ET-phen was partially compensated by D-serine[60].

**Stereotaxic application of D-serine into brainstem.** For stereotactic injections, CF1 mice (3–4 months old, $n = 6$) were deeply anesthetized by inhalation of iso-fluorane (3%) and their ears fixed with Stoelting dual-sided mouse ear bars and mounted in a Stoelting stereotaxic device model 51615 (Stoelting, Wood Dale, IL). After shaving the scalp fur, the skull was exposed through a midline skin incision to visualize lambda and bregma, this latter the reference stereotaxic coordinate (0,0). A tiny burr hole drilled through the skull with a dremel drill driving a 0.75 mm carbide bit was done in −7.32 mm anterior–posterior, 0 mm medial–lateral. Then the tip of a 4.2 mm guide cannula (PlasticOne, Roanoke, VA) was implanted through the hole to target the coordinates −7.32 mm anterior–posterior, 0 medial–lateral, 5.2 dorsal–ventral from dura, corresponding to the raphe area. The guide cannula was fixed to the surface of the skull using acrylic dental material and 3 screws were fixed to the bone. Once acrylics dried, the skin was sutured around the guide cannula and the 5.2 mm dummy cannula was inserted. The mice were allowed to recover 4 days before experiments and were treated with enrofloxacin (5 mg kg$^{-1}$ every 12 h) for avoid infections, and ketoprofen (3 mg kg$^{-1}$ every 12 h) for pain control.

With the mice in unrestrained conditions, the dummy cannula was replaced by a 5.2 mm long internal cannula (PlasticOne, Roanoke, VA) connected to a 5 µl Hamilton syringe (Hamilton company, Reno, NV). Using an analog device for controlling the advance of the syringe plunger, 0.2 µl of either saline or D-serine (30–300 µM) were injected into the raphe nucleus in 3 s. The internal cannula was maintained inside the guide cannula during 1 min up to be replaced by the dummy cannula. Before and after injection, ventilation was recorded in awake and free-moving mice using whole-body plethysmography. To define that the effects of the injections were consistent, injections were repeated in a particular mouse at least three times. At the end of the experiments, an injection of 0.5 µl methylene blue 0.1% in aCSF was performed for further localization of the injection site.

**In vitro experiments.** The isolated brainstem-spinal cord en bloc preparation in 0–4-day-old CF1 mice preserves the complete medullary respiratory network, whereas the medullary brainstem slice preparation, which contains the RN, NTS, and VRC, does not contain the RTN. In the en bloc preparations, the spontaneous inspiratory burst is recorded from the C3–C5 ventral roots with suction electrodes, whereas in the medullary slice preparation, the spontaneous inspiratory burst is recorded from brainstem respiratory neurons by placing the tip of the electrode on the VRC.

**En bloc preparation.** The procedures were similar to those described elsewhere[61]. In brief, mouse neonates from P0 to P3 were deeply anesthetized by isoflurane inhalation (Aesica Queenborough Limited, Kent, UK) and decapitated. The CNS was obtained through a dorsal approach under stereomicroscopy and immersed in cooled (4 °C) artificial cerebrospinal fluid (aCSF) containing (in mM): 125.0 NaCl, 3.0 KCl, 24.0 NaHCO₃, 1.25 KH₂PO₄, 1.0 CaCl₂, 1.25 MgSO₄•7H₂O (Sigma Aldrich, MO, USA), and 30.0 D-glucose (Merck) and equilibrated with O₂/CO₂ = 95/5% (pH 7.4). Preparations were decerebrated by a ponto-mesencephalic transection, and the cerebellum was removed. Isolated brainstem-spinal cord preparations were transferred to a 2 ml recording chamber, where they were superfused with aCSF (2 ml min$^{-1}$) at 25–26 °C.

**Slice preparation.** Neonatal P0–P4 medullary brainstem slices were obtained following the same initial steps as those used for en bloc preparations[62]. After the cerebellum was removed, the brainstem-spinal cord preparation was glued to an agar block with the rostral end up, and the block was mounted in a Vibroslice NVSL (World Precision Instruments, Sarasota, FL, USA). The brainstem was serially sectioned from the rostral to the caudal area in 200 µm steps until the facial nuclei were observed. Then, a section of 500 µm was discarded, and the next 700 µm slice was collected and immersed for 30 min in aCSF with a high concentration of K⁺ (9 mM) to improve the robustness and stability of the respiratory rhythm. Finally, the slice was transferred to a 250 µl recording chamber (RC-26G model, Warner Instruments, Hamden, CT, USA) mounted on the stage of an Eclipse FN1 microscope (Nikon, Tokyo, Japan) and superfused with aCSF containing 5 mM K⁺ (2 ml min$^{-1}$) at 29–30 °C.

The bath temperature was monitored with a thermistor probe placed at the outflow end of the en bloc or slice chamber and maintained at the desired temperature with an in-line heater operated by a temperature controller (TC-324C, Warner Instruments, Hamden, CT, USA).

**Electrical recording.** The spontaneous inspiratory burst activity was recorded, with a borosilicate glass suction electrode filled with aCSF, from C3 to C5 ventral roots at 26 ± 1 °C in en bloc and from the VRC at 29 ± 1 °C in medullary brainstem slices. The electrical signals were amplified by a low-noise differential amplifier (P55 A.C., Grass Instrument Co.), full-wave rectified and integrated (time constant 100 ms). Display, analysis, and saving of signals were made using Digipack 1320 AD acquisition system controlled with Axoscope 9.0 software (Axon Instruments, Union City, CA, USA).

**Hypercapnic acidosis.** Hypercapnic acidosis was generated by switching the aCSF gassing from 5% CO₂ to 10% CO₂ equilibrated in oxygen. This switch in bath gassing reduced the pH from 7.4 to 7.2. A typical hypercapnic acidosis challenge began with the superfusion of the preparations with aCSF at pH 7.4 for 5 min, followed by a superfusion with aCSF at pH 7.2 for 10 min, and a period of recovery at pH 7.4 for 10 min.

**Superfusion of substances during in vitro recordings.** Substances were dissolved in aCSF to the following final solution concentrations: 0.01–100 µM D-serine (Sigma Aldrich, St Louis, MO, USA), 10 µM dizocilpine, which is (+)-5-methyl-10,11-dihydro-5H-dibenzo[a,d] cyclohepten-5,10-iminemaleate (MK-801, Sigma Aldrich, St Louis, MO, USA), an NMDAR antagonist, 50 µM phenazine metho-sulfate (Sigma Aldrich, St Louis, MO, USA) and 50 µM phenazine ethosulfate (Sigma Aldrich, St Louis, MO, USA), both inhibitors of the serine racemase[56–58, 60], 0.1 U ml$^{-1}$ D-amino-acid oxidase (DAAO, Calzyme Laboratories Inc., San Luis Obispo, CA, USA) administered together with 260 U ml$^{-1}$ catalase (Calzyme Laboratories Inc., San Luis Obispo, CA, USA) for degrading D-serine and reducing peroxidase formation, respectively.

**Localization of D-serine-sensitive regions.** D-serine-sensitive regions were assessed by the microinjection of D-serine onto regions of interest at the caudal surface of medullary slices. Regions of interest included the NTS, RN, and VRC (preBötzinger complex level). In addition, the effects of D-serine were evaluated in the 5-SP and intermediate areas. The micropipette (0.75 mm ID) was filled with D-serine (300 µM in aCSF) and connected to the ejection port of a pneumatic picopump (model PV830, WPI, Sarasota, USA). The tip (3.75 µm ID) of the glass micropipette was placed on a region of interest at the caudal surface of medullary slices with a Narishige YOU-2 micromanipulator under stereomicroscopy. Pressure-ejected pulses (10 s, 3 psi) delivered a bolus of ~9 nl of solution into areas covering a region of 350 µm in diameter on the surface of the slice.

**Primary cell cultures.** CF1 mice (2 days old) were anesthetized with 3% isoflurane (Aesica Queenborough Limited, Kent, UK) and decapitated. Each independent primary astrocyte culture was prepared from the pooled brainstems of two mice or one cortex. Independent experiments were performed with five astrocyte cultures from the brainstem and three cortical cultures prepared from brain neocortex. The tissue was immersed in Hank's balanced salt solution (HBSS, Gibco, Thermo Fisher Scientific Inc., Waltham, MA USA), pH 7.4 at 4 °C, cleaned of meningeal and vascular tissue, and washed three times with sterile cold HBSS. Tissues were sectioned into small pieces and were digested in 1 ml of trypsin-EDTA (0.25% trypsin, 1 mM EDTA; Gibco, Thermo Fisher Scientific Inc., Waltham, MA, USA) in Hank's buffer without Ca²⁺/Mg²⁺ at 37 °C for 15 min. Tissue digestion was terminated by the addition of 1 ml of fetal bovine serum (FBS, Hyclone Laboratories Inc., Logan, UT, USA), and the cells were then subjected to mechanical dispersion by gentle pipetting. Astrocytes were plated on 60 mm (Falcon, USA) tissue culture dishes in complete media (Dulbecco's modified Eagle's medium (DMEM)) with nutrient mixture F-12 (DMEM-F12 (1:1); Gibco, Thermo Fisher Scientific Inc., Waltham, MA USA) containing 10% (v/v) FBS, 15 mM NaHCO₃ (Sigma Aldrich, St. Louis, MO, USA), 10 U ml$^{-1}$ penicillin, and 10 µg ml$^{-1}$ streptomycin (Hyclone Laboratories Inc., Logan, UT, USA). Cells were maintained in culture at 37 °C in a humidified atmosphere of 5% CO₂ in air until reaching confluence (~7 days). The medium was replaced every 2 days. Cell types were identified using cell type-specific markers: rabbit anti-Iba1 (Wako Chemicals Inc., Richmond, VA, USA) for microglia, mouse anti-neurofilament 68 (Sigma Aldrich, St. Louis, MO, USA) for neurons, and mouse (Cell Signaling Technologies, Danvers, MA, USA) or rabbit (DAKO, Glostrup, Denmark) anti-GFAP for astrocytes. The purity of the astrocyte cultures was higher than 95% in cells with DAPI positive nuclei (Sigma Aldrich, St. Louis, MO, USA) or Hoechst 33258 (Sigma Aldrich, St. Louis, MO, USA).

**Immunofluorescence.** Coverslips containing astrocyte cultures were washed with PBS containing 1 mM Ca²⁺. Cells were fixed in 2% p-formaldehyde at 20 °C for 15 min, permeabilized with 0.2% Triton X-100 in PBS for 15 min, blocked with 10% goat serum in PBS, and incubated at 4 °C overnight with mouse anti-GFAP (1:200; DAKO, Glostrup, Denmark), with either rabbit polyclonal anti-D-serine (1:200;

Ab6472, Abcam, Cambridge, UK) or goat polyclonal anti-D-serine racemase (1:200; sc-5751, Santa Cruz Biotechnology Inc., Dallas, TX, USA). For D-serine detection, incubation with the rabbit polyclonal anti D-serine was performed in the presence of 10 μM L-serine-glutaraldehyde-BSA conjugate to block cross-reactivity against L-serine.

The cross-reactivity ratio of the anti-D-ser antibody (Abcam Ab6472, same GEMAC and MoBiTec antibody as used in ref. [15]) with another amino acid, including L-serine, is lower than 1/50,000. This antibody has been used in colorimetric and fluorescent IHF assays[15, 44, 54, 55, 63–66]. The goat polyclonal anti serine racemase antibody (Santa Cruz Biotechnology, A17 sc-5751) was raised against the N terminus (amino acids 1–50) of the protein and was used for serine racemase detection in astrocytes in CNS tissue[54] and neuronal cell cultures[67]. Western blot analysis revealed a single band of ~37 kDa, the expected molecular weight[64]. No signal was observed with the anti-serine racemase antibody when it was used for immunohistochemistry in brain sections from serine racemase knockout (SR-KO) mice[48].

Coverslips were washed in PBS and incubated with the corresponding secondary antibody (anti-rabbit Alexa 488, anti-rabbit Alexa 546or anti goat Alexa 546 (1:100; Molecular Probes, Eugene, OR, USA)) in blocking solution at room temperature for 2.5 h. Nuclei were stained with 0.1 μg ml⁻¹ Hoechst 33258 (Sigma B2883, Sigma Aldrich, St. Louis, MO, USA). Coverslips were washed in PBS and water and mounted in non-fluorescent mounting medium (DAKO, Glostrup, Denmark).

**Release experiments**. After the culture medium was replaced with 2 ml aCSF, medullary brainstem and cortical astrocytes were placed in an incubator and exposed sequentially to the following environments: 5% $CO_2$ in air at 37 °C for 75 min (basal conditions), 10% $CO_2$ in air at 37 °C for 30 min (hypercapnic acidosis), and 5% $CO_2$ in air at 37 °C for 30 min (recovery to basal conditions). Samples of 50 μl of aCSF, equivalent to the 2.5% of the total culture volume, were taken and replaced with equal volume of aCSF at the end of the basal and recovery periods, and at 5 and 30 min of hypercapnic acidosis. Release data were normalized with the protein content of cultures, which was measured with the bicinchoninic acid (BCA) assay.

The contribution of either extracellular calcium or gap junction hemichannels on both hypercapnia-induced and potassium-induced D-serine release was explored by incubation of medullary brainstem astrocytes with aCSF calcium-free, that is aCSF with 0 mM calcium, 2 mM $Mg^{2+}$, and 1 mM EGTA (Sigma Aldrich, St. Louis, MO, USA) or with aCSF containing 10 or 100 μM carbenoxolone (Sigma Aldrich, St. Louis, MO, USA) or 1 mM probenecid (ApexBio, Boston, MA, USA) or 2 μM bafilomycin A1 (Tocris Bioscience, Bristol, UK) or 50 μM brefeldin A (Tocris Bioscience, Bristol, UK). Hypercapnic challenge consisted in the maintenance of the cultures during 75 min under normocapnia (5% $CO_2$ equilibrated in air) followed by 30 min of hypercapnia (10% $CO_2$ equilibrated in air). In the case of the potassium challenge, aCSF was maintained in isohydric normocapnia with extracellular KCl concentration of 3 mM for 30 min followed by 15 min of 50 mM KCl. Samples were collected (50 μl) at the end of every period and the volume replaced with aCSF. Samples were stored at −20 °C and analyzed within 3–24 h.

**Metabolic inhibition of astrocytes**. We used 5 mM sodium fluoroacetate (Sigma Aldrich, St. Louis, MO, USA) for selective metabolic inhibition of astrocytes. Fluoroacetate is incorporated by astrocytes but not neurons. Fluoroacetate reacts with coenzyme A to form fluoroacetyl CoA, which can replace acetyl CoA in Krebs cycle to produce fluorocitrate, which binds and blocks aconitase, and hence impairs the oxidative metabolism and energy production of astrocytes[68–70].

It is known that superfusion of brainstem slices with fluoroacetate in the presence of glutamine depresses astrocytes but preserves the respiratory network activity[32]. To link astrocyte function with D-serine release, hypercapnic tests were performed in medullary brainstem slices superfused initially with aCSF supplemented with 1.5 mM glutamine for 30 min (basal condition) followed by the superfusion with glutamine-supplemented aCSF containing 5 mM fluoroacetate for additional 30 min. To evaluate whether D-serine was able to rescue the respiratory response to hypercapnia in presence of metabolic inhibition of astrocytes, 50 μM D-serine was added to the glutamine-supplemented aCSF containing fluoroacetate.

**D-serine release during metabolic inhibition of astrocytes**. Eighteen caudal medullary brainstem slices (700 μM width) from neonatal mice (P1–P4) were divided in six groups of three slices each one. Each group of slices was incubated in a container of 1.5 ml with glutamine (1.5 mM)-supplemented aCSF at isohydric normocapnia (equilibration with $O_2/CO_2 = 95/5%$, pH 7.4) for 10 min. At the end of this period, a hypercapnic test was performed for 10 min by switching the gassing of the glutamine-supplemented aCSF from $CO_2$ 5% to $CO_2$ 10% balanced in oxygen (hypercapnic acidosis, pH 7.2). After 10 min of recovery with glutamine-supplemented aCSF at isohydric normocapnia, 5 mM fluoroacetate was added to the incubation medium. Exposure to fluoroacetate was maintained at isohydric normocapnia for 30 min before to start with a second hypercapnic test for 10 min in the presence of fluoroacetate. Samples were collected (100 μl) at the end of both isohydric normocapnia periods and hypercapnic tests and the volume replaced

with glutamine-supplemented aCSF. Samples were stored at −20 °C and analyzed within 3–24 h with LC-MS/MS.

**LC-MS/MS**. D-serine, L-serine, and L-glutamate were identified simultaneously from samples of the aCSF taken at different time points using an LC-MS/MS system (1200 s, triplequad 6410, Agilent, Palo Alto, CA, USA) with electrospray ionization in the positive mode. The eluents were 0.1% formic acid (A) and ultrapure acetonitrile (B) in linear steps for 0–5 min (100% B), 5–33 min (47% B), 33–55 min (65% B), 55–60 min (0% B), 60–65 min (50% B), and 65–70 min (100% B) with 0.5 ml min⁻¹ as a flow rate. The stationary phase consisted of a Chirobiotic®-T column (150 × 21 mm, 5 μm pore size; Supelco, Sigma-Aldrich, St. Louis, MO, USA). The mass spectrometry acquisition parameters were 330 °C as the gas temperature, 45 psi as the nebulization pressure, and 4500 V as the ionization voltage. The analyses were performed in the multiple reaction monitoring (MRM) mode, using the parameters summarized in Supplementary Table 1. The standards were purchased from Sigma-Aldrich (St. Louis, MO, USA). The calibration curves were obtained from 9.5 nM to 4.75 μM, and the injected volume was 20 μl. The retention times were 30.3 min for D-serine, 32.1 min for L-serine, and 38.5 min for glutamate.

**Data analysis**. The amplitude of the inspiratory burst of action potentials was estimated as the difference between the peak value of the integrated signal and the value of the integrated activity immediately before the onset of the burst, and it was expressed in arbitrary units. The cycle duration was measured from the onset of one burst of action potentials to the onset of the next burst of action potentials. The instantaneous respiratory rhythm frequency (fR) was calculated as the reciprocal value of the cycle duration and expressed in Hz. Changes in respiratory frequency were expressed as either the percentage of the basal fR or the relative fR; the latter corresponds to the ratio between the fR at a specific time and the basal fR. Values were expressed as the mean ± standard error of the mean (SEM).

A different investigator to whom performed the tissue culture procedures performed the HPLC detection of gliotransmitters. Analysis of electrophysiological recordings, although done by the same person who performed the experiments, was blinded with respect to the experimental condition.

Statistics were performed using GraphPad Prism 5, except for the Conovan post hoc test that was performed with an open-source non-commercial software. Sample sizes were defined on the basis of previous studies and the requirements for specific two-tailed tests. Non-parametric two-tailed statistics were preferred and used where appropriate. The statistical significance of the changes in respiratory frequency induced by hypercapnia either in vivo or in vitro was evaluated with the Wilcoxon signed-rank test. The comparison of multiple correlated groups was performed with Friedman's test followed by the Conovan post hoc test. In multiple independent groups, a Kruskal–Wallis test followed by the Dunn's multicomparison post hoc test was performed. After determining normality with the D'Agostino–Pearson normality test, differences in time courses between control and stimulated groups with repetitions showing homogenous variance were analyzed with mixed ANOVA followed by the Bonferroni post hoc test. Correlations between the basal burst frequency and the magnitude of hypercapnic acidosis responses were evaluated using the Spearman rank correlation coefficient. The null hypothesis was rejected if $P < 0.05$.

**Data availability**. Data generated or analyzed during this study are included in this published article and its Supplementary Information Files. Raw data for figures and results that appear in the text were deposited in https://doi.org/10.6084/m9.figshare.5239831.v1. Any additional or raw data are available from the corresponding authors on reasonable request.

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

## Acknowledgements

The current study was supported by grants Fondo Nacional del Desarrollo de la Ciencia y Tecnología (FONDECYT) 1130874 and 1171434 (J.E.), FONDECYT 1131025 and 1171645 (R.v.B.), FONDECYT 1140189 (G.Z.), DICYT_USACH (J.E.), Comisión Nacional de Ciencia y Tecnología (CONICYT) #21140669 (M.J.O.) and #21120594 (S. B.-C.). Part of this work was submitted by S.B.-C. to the Universidad de Santiago de Chile (USACH) in partial fulfillment of the requirements for the Ph.D. in Neuroscience; M.J.O. is a Ph.D. student in the Ph.D. Program in Neuroscience_USACH.

## Author contributions

S.B.-C. contributed to the design, data acquisition, analysis, and draft of the electrophysiology and ventilatory work and to the data acquisition in the D-serine release experiments; M.J.O. performed the tissue culture of astrocytes and the data acquisition in D-serine release experiments; G.Z. supervised the LC-MS/MS detection experiments and reviewed the draft of the manuscript; R.C. participated in the acquisition, analysis, and interpretation of data in the LC-MS/MS experiments; I.L. participated in the analysis of results and drafting of the article; R.v.B. participated in the funding, immunohistochemistry, data analysis, and drafting the article; and J.E. participated in the design, supervision, and funding of the work, in the acquisition of electrophysiological and ventilatory data, and in the writing of all versions of the manuscript.

## Additional information

**Competing interests:** The authors declare no competing financial interests.

