## [Peer review file · Nature Communications]

Reviewers' comments:

Reviewer #1 (Remarks to the Author):

This is a very interesting study revealing novel data implicating D-serine, derived from brainstem astrocytes, as an important contributor to the respiratory network response to CO₂ (acidosis) challenge, and indeed to basal respiratory frequency by way of tonic D-serineric tone.

The magnitude of the ventilatory response to CO₂ challenge in conscious mice appears quite low. Is this a feature of this particular strain? It would be more common for ventilation to at least double in response to significant hypercapnic challenge. In intact animals, elevated CO₂ in air will activate carotid body chemoreceptors as well as central CO₂ chemoreceptors, which should result in at least additive effects. Although the in vitro studies confine discussion to brainstem chemoreception, whole body challenges and intraperitoneal injection in intact animals opens the possibility of peripheral mechanisms of action. Some discussion offering consideration (exclusion?) of the carotid body is warranted. A potentially interesting complementary series of experiments to consider would be to examine the ventilatory response to CO₂ in hyperoxia, which should eliminate the contribution of the carotid body, isolating the CO₂ challenge to the central chemoreceptors. Also, a graded CO₂ challenge, 2%, 4%, 6% etc would allow the investigators to look at sensitivity in addition to gross effects on the magnitude of the response. The authors are well placed to conduct this additional study, which would add more convincing data.

Whilst the in vitro slice and en bloc preparations complement the study, revealing data that is consistent with the data obtained in conscious mice by plethysmography, the juxtaposition of the two draws focus to a potentially important aspect not directly addressed in the discussion: namely, developmental plasticity in CO₂ chemoreception. Since in vitro studies were performed in preparations from P0-P4 animals, some discussion of neonatal vs adult chemoreception is warranted. Importantly, it seems, the data reveal that D-serine is a crucial component of CO₂ chemoreception across this timeline in mouse. The limitations (and strengths) of the in vitro approach also warrant a short comment. Reduced preparations in adult animals would have complemented the plethysmography study.

Immunocytochemistry in neonatal and adult brain slices would complement the elegant approach of the authors exploring D-serine and D-serine racemase expression in cultured astrocytes.

Minor:

State "% of control" consistently e.g. Lines 80/81; 90/91; 132...to avoid confusion with % change. Include mean +/- SE for all responses.

Line 162: "a component" of the central respiratory response.

Line 283. What is the flow rate to the plethysmograph chamber?

Reviewed by:

Ken O'Halloran

University College Cork, Ireland

Reviewer #2 (Remarks to the Author):

This study constitutes an important finding in the field of respiration. The authors demonstrate that D-serine- that is released by astrocytes and stimulates respiration in response to hypercapnia is novel. The finding is interesting for a broad readership, and may also have clinical implications. The interaction between astrocytes and neurons has also been demonstrated in the context of the central hypoxic response, and the present study complements this important research. However,

the present study is very novel, since prior studies implicated ATP and purinergic receptors in the communication between astrocytes and neurons and to the best of my knowledge the role of serine has been overlooked. The authors make a convincing case that serine and the ventilatory response is mediated by the NMDA receptor.

The study is elegant because it integrates different levels – the study shows in dissociated culture that astrocytes are indeed releasing serine, they use histochemistry and explore the physiological relationship in vitro and in vivo, using electrophysiological and pharmacological approaches. The authors also show that there are D-serine sensitive areas within the medullary plane relevant for respiratory rhythm generation.

The in vivo experiments complement the in vitro findings, and both experimental approaches are consistent with the authors' main conclusion. The paper is straight-forward and well written.

However, there is one major issue, which can probably be overcome with additional experiments. The conclusions from the in vivo experiments are based on intraperitoneal (IP) injections. This is a rather indirect approach and while the consequences of D-serine in vivo are very similar to those demonstrated in vitro, there is not necessarily a causal relationship. This weakness could be overcome using more targeted D-serine (or antagonist) injections into the brainstem regions, or of course using optogenetic manipulations. At the very least the authors need to address this caveat in their discussion.

Reviewer #3 (Remarks to the Author):

This manuscript describes the results of the experiments aimed to investigate the potential role of D-serine released by astrocytes of the caudal brainstem in the mechanisms underlying central respiratory chemosensitivity. Experiments were performed using various models including cell culture, rhythmic brainstem slice preparations and conscious mice. The data obtained by the authors support the conclusions reached and suggest that release of D-serine by brainstem astrocytes contributes to the respiratory responses induced by systemic hypercapnia. This observation is very important for our understanding of the mechanisms underlying homeostatic regulation of blood and brain pH. The experiments described appear to be well performed, yet the brevity of the data reported begs additional experiments to clarify some points prior to publication. Any of the following additional experiments could be used to strengthen the manuscript:

1) CO₂-induced D-serine release by cultured astrocytes. The study would greatly benefit from identification of the potential mechanisms underlying D-serine release in response to CO₂. At the very least, the authors should investigate the role of connexin/pannexin hemichannels (see PMID: 20736421) and vesicular release mechanisms (PMID: 20647426). The robust effect of KCL may suggest a plausible (alternative) mechanism.

2) The strength of the conclusions reached by the authors depends on the use of pharmacological agents applied to inhibit D-serine racemase (such as phenazine methosulfate) or promote degradation of D-Serine (such as DAAO). Most of the agents used are not very specific. The only way to provide a definitive evidence in favour or against the authors' hypothesis would be to generate transgenic animals with conditional deletion of D-serine racemase in astrocytes and determining their respiratory responses to CO₂.

Other comments:

1) Figure 3 – please provide original traces to illustrate the effect of CO₂/acidification on the respiratory frequency recorded in rhythmic brainstem slices.

2) Accuracy of citations: For first (and only) direct demonstration of ATP release see PMID: 16001070, evidence that ATP activated RTN neurones - PMID: 2064742; mechanisms of release (could be similar for D-serine) involving connexin hemichannels - PMID: 20736421; prior evidence that brainstem and cortical astrocytes are different in terms of their Co₂/pH sensitivity + evidence of vesicular release recruitment - PMID: 23303924.

Answer to reviewers' comments

Reviewer's questions and comments are preceded with a numbered "C" and the text bolded and indented; our answers are preceded with a numbered "R". The text describing the main actions implying new experiments and the way they were inserted in the new version is highlighted in yellow, and references cited in response to the reviewers' comments are at the end of the document.

Reviewer #1:

C1. This is a very interesting study revealing novel data implicating D-serine, derived from brainstem astrocytes, as an important contributor to the respiratory network response to CO₂ (acidosis) challenge, and indeed to basal respiratory frequency by way of tonic D-serinergeric tone.

R1. We thank the reviewer for this positive comment.

C2. The magnitude of the ventilatory response to CO₂ challenge in conscious mice appears quite low. Is this a feature of this particular strain? It would be more common for ventilation to at least double in response to significant hypercapnic challenge.

R2. We agree with the reviewer that, in general, the changes in ventilation at least double in response to strong hypercapnic challenge. We think this reviewer's comment was based on the magnitude of change in fR and tidal volume (VT) we reported for controls in the previous version (around 130-150% of basal values). The new version incorporates in the line 157 (results) the hypercapnia-induced increase in minute volume ($V_E = V_T \times fR$), which was $206.7\% \pm 9.3\%$ of basal values. In addition, it is true that the magnitude of the ventilatory response to hypercapnia obtained in C57BL/6 background mice is more robust than that of the CD1 strain, and most of the plethysmography studies in the literature have been done in C57BL/6 mice. In the table below we have compared the magnitude of change, expressed as percentage of the basal, of the minute volume (VE), respiratory frequency (fR), and tidal volume (VT) in response to hypercapnia challenges of 5-10% CO₂ in air, in adult conscious mice. Comparison with the data shown in the table indicates that the increase in V_E, fR, and VT are closer to those reported by others.

	VE (% of basal)	fR (% of basal)	VT (% of basal)	reference
C57BL/6 10%	307	165	175	¹
Tg2 wt C57BL/6	252	140	150	²
Tg6 wt 5% CO ₂ C57BL/6	293	140	185	²
C57BL/6 5% strain	280	160	183	³

C3. In intact animals, elevated CO₂ in air will activate carotid body chemoreceptors as well as central CO₂ chemoreceptors, which should result in

at least additive effects. Although the in vitro studies confine discussion to brainstem chemoreception, whole body challenges and intraperitoneal injection in intact animals opens the possibility of peripheral mechanisms of action. Some discussion offering consideration (exclusion?) of the carotid body is warranted. A potentially interesting complementary series of experiments to consider would be to examine the ventilatory response to CO₂ in hyperoxia, which should eliminate the contribution of the carotid body, isolating the CO₂ challenge to the central chemoreceptors. Also, a graded CO₂ challenge, 2%, 4%, 6% etc would allow the investigators to look at sensitivity in addition to gross effects on the magnitude of the response. The authors are well placed to conduct this additional study, which would add more convincing data.

- R3.** We agree with the reviewer 1 that intraperitoneal injection of D-serine can activate peripheral mechanisms contributing to the ventilatory response. This point was also addressed by reviewer 2. The existence of D-serine and D-serine racemase have still not been demonstrated in the main peripheral chemoreceptors, the carotid bodies. However, carotid bodies contain glutamate NMDA receptors⁴, and NMDA receptor blockade with MK-801 attenuates the peripheral chemoreceptor responses induced by hypoxia, but not those induced by hypercapnia⁵. Thus, theoretically, when we applied i.p. D-serine, it could act on these peripheral NMDA receptors.

The experimental approach suggested by reviewer 1: to depress peripheral chemoreceptors with hyperoxia while an hypercapnic test is performed, in our context has several disadvantages: 1) The magnitude of the ventilatory effects induced by activation of central chemoreceptors depend on the peripheral chemoreceptor input, that is, the full expression of central chemoreception depends on the peripheral chemoafferent activity. For example, in conscious dogs, silencing of carotid bodies activity with hyperoxic, hypocapnic perfusate reduced the gain of the ventilatory response triggered by hypercapnic activation of central chemoreceptors by 81%^{6,7}. That means that in a situation of hyperoxia, the ventilatory response induced by direct central chemoreceptor stimulation will be greatly attenuated; 2) Hyperoxia generates hypoventilation which is opposite to the hyperventilation induced by hypercapnia and, in addition, hyperoxia will increase the arterial PCO₂, making the magnitude of hypercapnic stimulation difficult to control.

As suggested also by the editor, to demonstrate direct brainstem effects of D-serine in conscious adult mice, excluding peripheral mechanisms, appeared to be the main goal to solve. For that reason, we performed stereotaxic injection of D-serine into the brainstem throughout a guide cannula, implanted 4 days before, while whole-body plethysmography recordings were performed. We choose to test D-serine injected into raphe nucleus, because in our slice experiments the injection of D-serine in the raphe produced the strongest effects on respiratory frequency. We found that D-serine, but not saline injected into raphe nucleus increased the respiratory frequency in adult conscious mice.

We introduced these results and the respective methodology in lines 172 to 196 (results section) and 444 to 467 (methods). We generated a new Figure 4, in which the subfigures d-h are focused on these experiments. Respective changes in the

legend were also done.

C4. Whilst the *in vitro* slice and *en bloc* preparations complement the study, revealing data that is consistent with the data obtained in conscious mice by plethysmography, the juxtaposition of the two draws focus to a potentially important aspect not directly addressed in the discussion: namely, developmental plasticity in CO₂ chemoreception. Since *in vitro* studies were performed in preparations from P0-P4 animals, some discussion of neonatal vs adult chemoreception is warranted. Importantly, it seems, the data reveal that D-serine is a crucial component of CO₂ chemoreception across this timeline in mouse. The limitations (and strengths) of the *in vitro* approach also warrant a short comment. Reduced preparations in adult animals would have complemented the plethysmography study.

R4: We agree with the reviewer that developmental plasticity in CO₂ chemoreception is an important issue, which we want to explore in future research. As expressed by the reviewer, our data reveal that D-serine is contributing to the ventilatory response in neonates and adults. Slices and *en bloc* approaches allow a reliable control of the superfusion medium and an easy access to specific components of the neural network. However, due to unknown reasons, is not possible to obtain rhythmic and reliable slice preparations from adult mice and it is impossible to obtain rhythmic *en bloc* preparations from neonates older than P4. For that reason we restrict these approaches to mouse neonates. On the other hand, to show that D-serine mechanisms can be observed in conscious adult animals we performed experiments using plethysmography. We agree with the reviewer that, theoretically, complementary experiments in reduced (*in vitro*) preparations obtained from adult mice might help to compare D-serine contribution to hypercapnia in neonates and adults. However, the reviewer's suggestion of complementary experiments in reduced (*in vitro*) preparations in adults is technically impracticable.

As requested by the reviewer, in the new version of the manuscript we discuss about neonatal vs adult chemoreception centered in the raphe contribution to central chemoreception in lines 258-271 (Discussion section). In addition, in the same paragraph we addressed the limitations (and strengths) of the *in vitro* approach under the perspective of comparison of neonate vs. adult contribution to central chemoreception.

C5. Immunocytochemistry in neonatal and adult brain slices would complement the elegant approach of the authors exploring D-serine and D-serine racemase expression in cultured astrocytes.

R5 We found that immunohistochemistry explorations in neonatal vs. adult mouse brain slices should be part of a more systematic study of developmental changes in the contribution of D-serine to CO₂ chemoreception which we want to perform in the future.

Minor:

C6. State "% of control" consistently e.g. Lines 80/81; 90/91; 132...to avoid

- confusion with % change. Include mean +/- SE for all responses.**
- R6.** We included mean \pm SEM in all the expressions along the manuscript. In the new version we avoid the confusion between % change and % of control. We preferentially and consistently describe the values obtained during an experimental condition as percentage respect of basal value. See lines 89, 116, 128, 137, 158.
- C7.** Line 162: "a component" of the central respiratory response.
- R7** We replaced "mediator" by the suggested expression "a component" in the first line of discussion (line 199).
- C8. Line 283. What is the flow rate to the plethysmograph chamber?**
- R8.** The chamber was continuously supplied with gas mixtures at 180 ml/min. We introduced this antecedent in the section method, line 401.

Reviewer #2:

- C1. This study constitutes an important finding in the field of respiration. The authors demonstrate that D-serine- that is released by astrocytes and stimulates respiration in response to hypercapnia is novel. The finding is interesting for a broad readership, and may also have clinical implications. The interaction between astrocytes and neurons has also been demonstrated in the context of the central hypoxic response, and the present study complements this important research. However, the present study is very novel, since prior studies implicated ATP and purinergic receptors in the communication between astrocytes and neurons and to the best of my knowledge the role of serine has been overlooked. The authors make a convincing case that serine and the ventilatory response is mediated by the NMDA receptor.**

The study is elegant because it integrates different levels – the study shows in dissociated culture that astrocytes are indeed releasing serine, they use histochemistry and explore the physiological relationship in vitro and in vivo, using electrophysiological and pharmacological approaches. The authors also show that there are D-serine sensitive areas within the medullary plane relevant for respiratory rhythm generation.

The in vivo experiments complement the in vitro findings, and both experimental approaches are consistent with the authors' main conclusion. The paper is straight-forward and well written.

- R1** We appreciate very much the positive comments of this reviewer
- C2. However, there is one major issue, which can probably be overcome with additional experiments. The conclusions from the in vivo experiments are based on intraperitoneal (IP) injections. This is a rather indirect approach and while the consequences of D-serine in vivo are very similar to those demonstrated in vitro, there is not necessarily a causal relationship. This weakness could be overcome using more targeted D-serine (or antagonist) injections into the**

brainstem regions, or of course using optogenetic manipulations. At the very least the authors need to address this caveat in their discussion.

- R2. This point was also raised by the reviewer 1, and we agree that the intraperitoneal injection may activate peripheral mechanisms. The fact that the time courses of both the increase in the brain content of D-serine in rats and the respiratory effects in mice after a single i.p. injection of D-serine are similar is very suggestive of central effects. Additionally, our in vitro experiments show that in reduced deafferented preparations D-serine is able to exert respiratory effects. However, and we agree with this, a direct evidence of actions of D-serine on brainstem in adult conscious mice was required. Therefore, we decided to inject D-serine directly into the raphe nucleus by the reasons mentioned in the answer R3 for reviewer 1. As we mentioned there, we introduced these results and the respective methodology in lines 172 to 196 (results section) and 444 to 467 (methods). We generated a new Figure 4, in which the subfigures d-h are focused on these experiments. The respective legend was also modified.

Reviewer #3 (Remarks to the Author):

- C1 This manuscript describes the results of the experiments aimed to investigate the potential role of D-serine released by astrocytes of the caudal brainstem in the mechanisms underlying central respiratory chemosensitivity. Experiments were performed using various models including cell culture, rhythmic brainstem slice preparations and conscious mice. The data obtained by the authors support the conclusions reached and suggest that release of D-serine by brainstem astrocytes contributes to the respiratory responses induced by systemic hypercapnia.**

R1 We thank the positive comments of the reviewer.

- C2. This observation is very important for our understanding of the mechanisms underlying homeostatic regulation of blood and brain pH. The experiments described appear to be well performed, yet the brevity of the data reported begs additional experiments to clarify some points prior to publication. Any of the following additional experiments could be used to strengthen the manuscript:**

1) CO₂-induced D-serine release by cultured astrocytes. The study would greatly benefit from identification of the potential mechanisms underlying D-serine release in response to CO₂. At the very least, the authors should investigate the role of connexin/pannexin hemichannels (see PMID: 20736421) and vesicular release mechanisms (PMID: 20647426). The robust effect of KCL may suggest a plausible (alternative) mechanism.

2) The strength of the conclusions reached by the authors depends on the use of pharmacological agents applied to inhibit D-serine racemase (such as phenazine methosulfate) or promote degradation of D-Serine (such as DAAO). Most of the agents used are not very specific. The only way to provide a definitive evidence in favour or against the authors' hypothesis would be to generate transgenic

animals with conditional deletion of D-serine racemase in astrocytes and determining their respiratory responses to CO₂.

R2. Point 1: We included experiments to get insight into how astrocytes respond to elevated CO₂ to release D-serine evaluating the contribution of either extracellular calcium or gap junction hemichannels. Medullary brainstem astrocytes were incubated with aCSF calcium-free (that means aCSF containing 0 Ca²⁺, 2 mM Mg²⁺, and 1mM EGTA) or with aCSF containing carbenoxolone (100µM) or probenecid (1mM) for 30 min under normocapnia (5% CO₂ equilibrated in air) followed by 30 min of hypercapnia (10% CO₂ equilibrated in air). Carbenoxolone, at the dose used here, blocks both connexin and pannexin-1 hemichannels, while probenecid selectively blocks pannexin-1 hemichannel. Incubation with calcium-free aCSF elevated D-serine baseline, which is suggestive of gate opening of connexin and pannexin in calcium-free medium. Carbenoxolone (100µM) or probenecid (1mM) abolished the hypercapnia-induced release of D-serine, while this persisted, although reduced, in calcium-free aCSF. Similar results were obtained with high potassium-evoked D-serine release, which is also suggestive of pannexin-1 involvement.

In the new version of the manuscript we introduced the description of these experiments in results, lines 71-82, in methods, lines 594-603, we generated a new subfigure and respective legend (Fig 1h). In addition, we addressed these results in the discussion, lines 215 to 227.

R2. Point 2. We agree with the reviewer 3 that the use of pharmacological agents applied to inhibit D-serine racemase (such as phenazine methosulfate) or promote degradation of D-Serine (such as DAAO) are not very specific. For that reason we used different and complementary approaches to address this problem. However, we do not agree with the notion that to provide a definitive evidence we need to generate transgenic animals with conditional deletion of D-serine racemase in astrocytes and determine their respiratory responses to hypercapnia. Although this strategy, at a first glance, looks impeccable and highly recommendable, we think it will most likely yield uncertain results and no definitive evidence. The reasons are the followings:

- a) There is only one work with conditional deletion of D-serine racemase (SR)⁸. In this work, tamoxifen-inducible GFAP⁺ astrocyte-specific Cre-recombinase-mediated SR knockout reduced the expression of this enzyme only by 15% compared with its control values in the mouse forebrain, the site of the brain with the highest SR levels⁸. The small decrease in D-serine racemase resulted in the absence of differences in both the D-serine levels and the hippocampal LTP induction⁸. It should be noted that LTP induction is known to be dependent on astrocyte release of D-serine^{9,10}. In contrast, the tamoxifen-induced selective deletion of neuronal SR resulted in a 60% reduction of this enzyme levels in cerebral cortex and hippocampus and was associated to hippocampal LTP impairment⁸. Then, affecting specifically the synthesis of D-serine in astrocytes may be not enough to reveal astrocyte function, for instance, in LTP.
- b) In contrast, the above results can be explained on terms of the current notion

that proposes that neurons have a greater content of SR and synthesize more D-serine than astrocytes⁸. However, astrocytes, but not neurons are able of synthesizing L-serine, the precursor for the D-serine synthesis. In addition, in contrast to neurons, astrocytes are able of storing D-serine. Thus, a major proportion of the D-serine stored by astrocytes appears to be synthesized by neurons¹¹. In fact, the “serine shuttle” hypothesis proposes that “*astrocytes synthesize and export L-serine required for the synthesis of D-serine by the predominantly neuronal D-serine racemase. D-Serine synthesized and released by neurons can be further taken up by astrocytes for storage and activity-dependent release*”¹¹.

Therefore, for the purpose of providing a definitive evidence, the proposed experiments using conditional deletion of D-serine in astrocytes do not preclude a clear result, because the levels of astrocytic D-serine might be maintained by brainstem neurons.

- c) Other point is crucial with conditional transgenics. The time course to reduce the levels of D-serine racemase in conditional transgenic are in the order of 10 and 15 days after the last injection of tamoxifen⁸. Such a relative long time most probable will produce metabolic alterations, because D-serine racemase not only catalyzes the production of D-serine from L-serine, but also catalyzes the production of pyruvate from L-serine by an α,β -elimination reaction¹². In fact, D-serine racemase produces 3 molecules of pyruvate for each molecule of D-serine in presence of the cofactors Mg^{2+} and ATP¹². Pyruvate can be used by the mitochondrial tricarboxylic acid (TCA) cycle, which can lead to the production of diverse amino acids such as glutamate and GABA, among others. Pyruvate can be converted into lactate, thus providing energy for neurons. Furthermore, it can be also used by pyruvate carboxylase, a specific glial enzyme, to form aspartate¹³. Therefore, D-serine racemase is closely linked to energy metabolism of astrocytes and neurons. Thus, the effects in glial and neuronal metabolism may be important. Future studies should be done to determine whether D-serine racemase-derived pyruvate has an important metabolic role¹¹.

By contrast, our pharmacological approach using SR blockers achieve a transient reduction of the D-serine racemase activity *in vivo*, approximately 2 hrs after i.p. injection, with total recovery after 24 h of a single injection. The delay is shorter in slices, being about 45 min and the total recovery is obtained after 2 h of washout. **That means that our approach, in contrast to the use of a conditional SR knockout mice, minimizes the associated metabolic effects and it gives few time to the emergence of homeostatic plastic changes that could counterbalance the functional deficits cauded by deletion of D-serine racemase.**

- d) Constitutive inactivation of the D-serine racemase gene (constitutive SR KO) indicates that this enzyme contributes to 80-90% of D-serine biosynthesis in the mouse neocortex (frontal cortex), striatum and hippocampus¹⁴⁻¹⁶, but it almost does not affect D-serine levels in the cerebellum. These results suggest first that there are other sources responsible of the 10-20% of D-serine at the brain, such as glycine cleavage system, phosphoserine phosphatase, diet, gastrointestinal bacteria¹⁴, or a yet unknown pathway¹⁶. Second, these results indicate that the

effect of the deletion of D-serine racemase expression on the magnitude of the change in D-serine levels is not equivalent in different brain regions. **In fact, the behavior of D-serine in the brainstem is not known yet.**

In summary, although we agree with reviewer 3 that the agents used to decrease the levels of D-serine (D-serine degradation with DAAO or D-serine synthesis inhibition with phenazine methosulfate) are not very specific for D-serine when analyzed separately, we consider that the strength of our approach resides in the coherence of the results taken as a whole, the short delay to reduce D-serine racemase activity or the levels of extracellular D-serine, implying a minimal effect on astrocytic and neuronal metabolism. By contrast, a conditional astrocytic SR knockout will not yield a definitive result by the above mentioned reasons.

Nevertheless, to provide a more direct evidence that astrocytes are the source of D-serine that drives breathing in response to elevated CO₂, we performed a new set of experiments using metabolic specific manipulation of astrocytes in medullary brainstem slices. We used fluoroacetate, a known blocker of the Krebs cycle in astrocytes, and determined its effects on the respiratory response to hypercapnia and D-serine release. We found that a metabolic inhibition of astrocytes impaired the hypercapnia-induced increase in fR and reduced the release of D-serine from slices. Moreover, application of exogenous D-serine was enough to restore the respiratory response to hypercapnia. We introduced these experiments in results (lines 172 to 196), methods (lines 605 to 616) , we generate a new Figure 6, and in discussion section (lines 253 to 257). In addition, we incorporate a discussion of the L-serine shuttle and the role of astrocytes and neurons on the shuttle mechanism (lines 234 to 253).

Other comments:

- C3. Figure 3 – please provide original traces to illustrate the effect of CO₂/acidification on the respiratory frequency recorded in rhythmic brainstem slices.**
- R3** In the new version we provided original traces to illustrate the changes in fR induced by hypercapnia in vitro (new Figure 4 replaces former Figure 3).
- C4. Accuracy of citations: For first (and only) direct demonstration of ATP release see PMID: 16001070, evidence that ATP activated RTN neurones - PMID: 2064742; mechanisms of release (could be similar for D-serine) involving connexin hemichannels - PMID: 20736421; prior evidence that brainstem and cortical astrocytes are different in terms of their Co₂/pH sensitivity + evidence of vesicular release recruitment - PMID: 23303924.**
- R4.** We thank the reviewer for this clarification. In the new version of the manuscript we have cited all this relevant papers with accuracy, including the introduction section.

References cited in the answers to reviewer's comments

- 1 Hodges, M. R. & Richerson, G. B. Interaction between defects in ventilatory and thermoregulatory control in mice lacking 5-HT neurons. *Respiratory physiology & neurobiology* **164**, 350-357, doi:10.1016/j.resp.2008.08.003 (2008).
- 2 Laouafa, S. *et al.* Hypercapnic ventilatory response is decreased in a mouse model of excessive erythrocytosis. *American journal of physiology. Regulatory, integrative and comparative physiology* **311**, R940-R947, doi:10.1152/ajpregu.00226.2016 (2016).
- 3 Li, A. & Nattie, E. Serotonin transporter knockout mice have a reduced ventilatory response to hypercapnia (predominantly in males) but not to hypoxia. *The Journal of physiology* **586**, 2321-2329, doi:10.1113/jphysiol.2008.152231 (2008).
- 4 Liu, Y. *et al.* Exposure to cyclic intermittent hypoxia increases expression of functional NMDA receptors in the rat carotid body. *J Appl Physiol (1985)* **106**, 259-267, doi:10.1152/jappphysiol.90626.2008 (2009).
- 5 Ohtake, P. J., Torres, J. E., Gozal, Y. M., Graff, G. R. & Gozal, D. NMDA receptors mediate peripheral chemoreceptor afferent input in the conscious rat. *J Appl Physiol (1985)* **84**, 853-861 (1998).
- 6 Blain, G. M., Smith, C. A., Henderson, K. S. & Dempsey, J. A. Peripheral chemoreceptors determine the respiratory sensitivity of central chemoreceptors to CO₂. *The Journal of physiology* **588**, 2455-2471, doi:10.1113/jphysiol.2010.187211 (2010).
- 7 Smith, C. A., Blain, G. M., Henderson, K. S. & Dempsey, J. A. Peripheral chemoreceptors determine the respiratory sensitivity of central chemoreceptors to CO₂ : role of carotid body CO₂. *The Journal of physiology* **593**, 4225-4243, doi:10.1113/JP270114 (2015).
- 8 Benneyworth, M. A., Li, Y., Basu, A. C., Bolshakov, V. Y. & Coyle, J. T. Cell selective conditional null mutations of serine racemase demonstrate a predominate localization in cortical glutamatergic neurons. *Cellular and molecular neurobiology* **32**, 613-624, doi:10.1007/s10571-012-9808-4 (2012).
- 9 Han, H., Peng, Y. & Dong, Z. D-Serine rescues the deficits of hippocampal long-term potentiation and learning and memory induced by sodium fluoroacetate. *Pharmacology, biochemistry, and behavior* **133**, 51-56, doi:10.1016/j.pbb.2015.03.017 (2015).
- 10 Yang, Y. *et al.* Contribution of astrocytes to hippocampal long-term potentiation through release of D-serine. *Proceedings of the National Academy of Sciences of the United States of America* **100**, 15194-15199, doi:10.1073/pnas.2431073100 (2003).
- 11 Wolosker, H. Serine racemase and the serine shuttle between neurons and astrocytes. *Biochimica et biophysica acta* **1814**, 1558-1566, doi:10.1016/j.bbapap.2011.01.001 (2011).
- 12 De Miranda, J., Panizzutti, R., Foltyn, V. N. & Wolosker, H. Cofactors of serine racemase that physiologically stimulate the synthesis of the N-methyl-D-aspartate (NMDA) receptor coagonist D-serine. *Proceedings of the National Academy of Sciences of the United States of America* **99**, 14542-14547, doi:10.1073/pnas.222421299 (2002).
- 13 Martineau, M., Baux, G. & Mothet, J. P. D-serine signalling in the brain: friend and foe. *Trends in neurosciences* **29**, 481-491, doi:10.1016/j.tins.2006.06.008 (2006).

- 14 Basu, A. C. *et al.* Targeted disruption of serine racemase affects glutamatergic neurotransmission and behavior. *Molecular psychiatry* **14**, 719-727, doi:10.1038/mp.2008.130 (2009).
- 15 Horio, M. *et al.* Levels of D-serine in the brain and peripheral organs of serine racemase (Srr) knock-out mice. *Neurochemistry international* **59**, 853-859, doi:10.1016/j.neuint.2011.08.017 (2011).
- 16 Inoue, R., Hashimoto, K., Harai, T. & Mori, H. NMDA- and beta-amyloid1-42-induced neurotoxicity is attenuated in serine racemase knock-out mice. *The Journal of neuroscience : the official journal of the Society for Neuroscience* **28**, 14486-14491, doi:10.1523/JNEUROSCI.5034-08.2008 (2008).

Reviewers' comments:

Reviewer #1 (Remarks to the Author):

The authors have adequately addressed the points that I raised in my initial review. The changes made to the manuscript have strengthened the study. My reference to the potential use of alternative reduced preparations in adult mice, related to the potential use of a working heart brainstem preparation, with access to brainstem sites for drug administration. The authors have addressed this issue by providing additional experiments which employ focal medullary applications in conscious mice illustrating the central respiratory effects of D-serine providing additional convincing evidence in support of the study.

I commend the authors on a very nice study.

Ken O'Halloran
University College Cork, Ireland.

Reviewer #2 (Remarks to the Author):

The authors have done a careful job in revising the manuscript. They addressed all my concerns. Specifically the concern that Serine acts on other targets. The discussion added increases the confidence that the effects are not due to other peripheral effects. Thus, I have no further comments.

Reviewer #3 (Remarks to the Author):

This revised manuscript describes the results of the experimental studies aimed to investigate the role of D-serine released by astrocytes of the caudal brainstem in the mechanisms underlying central respiratory chemosensitivity. As I have indicated in my previous review, the data presented are interesting and important for our understanding of the mechanisms underlying homeostatic regulation of blood and brain pH. The authors satisfactorily addressed most of my initial concerns. However, I am not convinced that the pharmacological analysis of CO₂-induced D-serine release performed by the authors is robust enough to conclude on the involvement of pannexin-1 as a conduit of D-serine release. The authors indicate in the text (page 2): "The label for D-serine was uniformly distributed in the cytoplasm of the majority of the astrocytes as fine puncta (Fig. 1a)...". This would suggest potential vesicular accumulation, yet the Figure 1a is provided in such a low resolution that "puncta" is not really visible. To support the potential role of pannexin-1 the reader would ideally like to see the effect of carbenoxolone applied in 10 μ M concentration (in this concentration carbenoxolone blocks pannexins, but not connexin hemichannels). Also, to determine the potential role of vesicular release mechanisms (suggested by the labelled "puncta") compounds like brefeldin A and bafilomycin can be applied in the experiments in cell culture.

The authors may also wish to discuss the results of two recent relevant papers published since their initial submission. One suggests that brainstem astrocytes are specialized as CO₂/pH sensors (PMID: 27798130). The other suggests that all CNS astrocytes are able to sense CO₂ and mediate the effect of CO₂ on cerebral vasculature (PMID: 28137973).

Reviewer's concerns and comments are preceded with a numbered "C" and the text bolded and indented; our answers are preceded with a numbered "R". The text describing the main actions and the way they were inserted in the new version is highlighted in yellow, and references cited in response to the reviewers' comments are at the end of the document.

Reviewer #1

C1. The authors have adequately addressed the points that I raised in my initial review. The changes made to the manuscript have strengthened the study. My reference to the potential use of alternative reduced preparations in adult mice, related to the potential use of a working heart brainstem preparation, with access to brainstem sites for drug administration. The authors have addressed this issue by providing additional experiments which employ focal medullary applications in conscious mice illustrating the central respiratory effects of D-serine providing additional convincing evidence in support of the study.

I commend the authors on a very nice study.

R1. We appreciate and thank very much these positive comments.

Reviewer #2

C1. The authors have done a careful job in revising the manuscript. They addressed all my concerns. Specifically the concern that Serine acts on other targets. The discussion added increases the confidence that the effects are not due to other peripheral effects.

Thus, I have no further comments.

R1 We also appreciate and thank very much the positive comments of this reviewer

Reviewer #3

C1 This revised manuscript describes the results of the experimental studies aimed to investigate the role of D-serine released by astrocytes of the caudal brainstem in the mechanisms underlying central respiratory chemosensitivity. As I have indicated in my previous review, the data presented are interesting and important for our understanding of the mechanisms underlying homeostatic regulation of blood and brain pH. The authors satisfactory addressed most of my initial concerns.

R1 We thank the positive comments of the reviewer.

C2. However, I am not convinced that the pharmacological analysis of CO₂-induced D-serine release performed by the authors is robust enough to conclude on the involvement of pannexin-1 as a conduit of D-serine release. The authors indicate in the text (page 2): "The label for D-serine was uniformly distributed in the cytoplasm of the majority of the astrocytes as fine puncta (Fig. 1a)...". This would suggest potential vesicular accumulation, yet the Figure 1a is provided in such a low resolution that "puncta" is not really visible.

To support the potential role of pannexin-1 the reader would ideally like to see the effect of carbenoxolone applied in 10 uM concentration (in this concentration carbenoxolone blocks pannexins, but not connexin hemichannels). Also, to determine the potential role of vesicular release mechanisms (suggested by the labelled "puncta") compounds like brefeldin A and bafilomycin can be applied in the experiments in cell culture.

R2. We performed the experiments suggested by this referee:

[2.1] Incubation of medullary brainstem astrocytes with aCSF containing 10µM carbenoxolone blocked the increase in D-serine concentration induced by hypercapnia. As well mentioned by the referee 3, 10µM carbenoxolone is known to block pannexin-1, but not connexin hemichannels. We reported in the previous version of the manuscript that 100 µM carbenoxolone (that blocks both pannexin-1 and connexins) and 1 mM probenecid (that is considered to block pannexin-1 and not connexin) also abolished the increase in D-serine induced by hypercapnia. Therefore, the new result confirms our proposition that pannexin-1 is involved in D-serine release from medullary astrocytes.

[2.2] Incubation of medullary brainstem astrocytes with aCSF containing 2 µM bafilomycin A1 or 50 µM brefeldin A blocked the increase in D-serine concentration induced by hypercapnia. These results suggest that vesicular release mechanisms are also involved.

In the new version of the manuscript we introduced the changes in abstract (line 25), results (line 94, lines 96-103, line 105, lines 108-110), discussion (line 246, lines 248-250, lines 257-263), methods (line 511, lines 512-513), legend fig. 1 (lines 765-771), and figure 1h. As expressed in lines 257-263 of discussion section, the impairment of vesicular mechanism with bafilomycin and brefeldin could represent an indirect effect mediated by the release of other gliotransmitter that could trigger D-serine release.

C3. **The authors may also wish to discuss the results of two recent relevant papers published since their initial submission. One suggests that brainstem astrocytes are specialized as CO₂/pH sensors (PMID: 27798130). The other suggests that all CNS astrocytes are able to sense CO₂ and mediate the effect of CO₂ on cerebral vasculature (PMID: 28137973).**

R3. We thank the referee by these suggestions. We included both references in the new version of the manuscript. PMID 28137973¹ is mentioned in Introduction (line 40) and Discussion (line 230) supporting the idea that cortical and brainstem astrocytes can sense H⁺/CO₂. This is a controversial issue, because other groups have reported that astrocytes from the cerebral cortex or dorsal brainstem are not able to generate [Ca²⁺]_i signals in response to acidification^{2,3}.

The other reference, PMID 27798130⁴ is included when we allude to astrocytic mechanisms to sense H⁺/CO₂ in the line 232 of discussion.

References in the answer to the reviewer's comments

- 1 Howarth, C. *et al.* A Critical Role for Astrocytes in Hypercapnic Vasodilation in Brain. *The Journal of neuroscience : the official journal of the Society for Neuroscience* **37**, 2403-2414, doi:10.1523/JNEUROSCI.0005-16.2016 (2017).
- 2 Kasymov, V. *et al.* Differential sensitivity of brainstem versus cortical astrocytes to changes in pH reveals functional regional specialization of astroglia. *The Journal of neuroscience : the official journal of the Society for Neuroscience* **33**, 435-441, doi:10.1523/JNEUROSCI.2813-12.2013 (2013).
- 3 Gourine, A. V. *et al.* Astrocytes control breathing through pH-dependent release of ATP. *Science* **329**, 571-575, doi:10.1126/science.1190721 (2010).
- 4 Turovsky, E. *et al.* Mechanisms of CO₂/H⁺ Sensitivity of Astrocytes. *The Journal of neuroscience : the official journal of the Society for Neuroscience* **36**, 10750-10758, doi:10.1523/JNEUROSCI.1281-16.2016 (2016).

REVIEWERS' COMMENTS:

Reviewer #3 (Remarks to the Author):

The authors satisfactorily addressed all my remaining concerns. The data presented in the revised manuscript are interesting and important for our understanding of the physiological mechanisms underlying homeostatic regulation of blood and brain pH. I have no further comments.

Answer to reviewers' comments NCOMMS-16-21368C
July 24th 2017

Reviewer's last concerns and comments are preceded with a numbered "C" and the text bolded and indented; our answer is preceded with a numbered "R".

Reviewer #3

C1 The authors satisfactorily addressed all my remaining concerns. The data presented in the revised manuscript are interesting and important for our understanding of the physiological mechanisms underlying homeostatic regulation of blood and brain pH. I have no further comments.

R1 We thank the positive comments of the reviewer and the suggestions to perform studies about the mechanisms of release of astrocytic D-serine. This allowed us to get a more complete insight of the phenomenon.